# Using Microcystin Gene Copies to Determine Potentially-Toxic Blooms, Example from a Shallow Eutrophic Lake Peipsi

**DOI:** 10.3390/toxins12040211

**Published:** 2020-03-26

**Authors:** Kristel Panksep, Marju Tamm, Evanthia Mantzouki, Anne Rantala-Ylinen, Reet Laugaste, Kaarina Sivonen, Olga Tammeorg, Veljo Kisand

**Affiliations:** 1Chair of Hydrobiology and Fishery, Institute of Agricultural and Environmental Sciences, Estonian University of Life Sciences, 51006 Tartu, Estonia; 2Department F.-A. Forel for Environmental and Aquatic Sciences, University of Geneva, 1205 Geneva, Switzerland; 3Institute of Biotechnology, University of Helsinki, 00790 Helsinki, Finland; 4Department of Microbiology, University of Helsinki, 00014 Helsinki, Finland; 5Ecosystems and Environmental Research Programme, University of Helsinki, 00014 Helsinki, Finland; 6Institute of Technology, University of Tartu, 50411 Tartu, Estonia

**Keywords:** cyanobacteria, qPCR, *mcyE*, microcystins, MC quota, Lake Peipsi

## Abstract

Global warming, paired with eutrophication processes, is shifting phytoplankton communities towards the dominance of bloom-forming and potentially toxic cyanobacteria. The ecosystems of shallow lakes are especially vulnerable to these changes. Traditional monitoring via microscopy is not able to quantify the dynamics of toxin-producing cyanobacteria on a proper spatio-temporal scale. Molecular tools are highly sensitive and can be useful as an early warning tool for lake managers. We quantified the potential microcystin (MC) producers in Lake Peipsi using microscopy and quantitative polymerase chain reaction (qPCR) and analysed the relationship between the abundance of the *mcyE* genes, MC concentration, MC variants and toxin quota per *mcyE* gene. We also linked environmental factors to the cyanobacteria community composition. In Lake Peipsi, we found rather moderate MC concentrations, but microcystins and microcystin-producing cyanobacteria were widespread across the lake. Nitrate (NO_3_^−^) was a main driver behind the cyanobacterial community at the beginning of the growing season, while in late summer it was primarily associated with the soluble reactive phosphorus (SRP) concentration. A positive relationship was found between the MC quota per *mcyE* gene and water temperature. The most abundant variant—MC-RR—was associated with MC quota per *mcyE* gene, while other MC variants did not show any significant impact.

## 1. Introduction

Eutrophication of aquatic systems caused by anthropogenic nutrient enrichment is a critical environmental problem of the 21st century [1,2]. Extensive research has shown that increased nutrient-loading shifts phytoplankton communities towards the dominance of bloom-forming and potentially toxic cyanobacteria [3,4,5]. However, many symptoms of eutrophication are affected and escalated by global warming, it was predicted that the extent and frequency of the harmful cyanobacterial blooms will increase in a warmer climate [6,7,8,9,10]. Anthropogenic pressures render shallow lake ecosystems especially vulnerable to environmental change and the subsequent boosting of cyanobacterial occurrence [11,12].

Cyanobacterial blooms pose a substantial health risk to humans and animal species due to the cyanotoxins they produce. These secondary metabolites are mainly stored inside the cyanobacterial cell and are released into the water during cell lysis, potentially leading to high toxin concentrations [13]. Aquatic life such as fish, crustaceans, mussels, and molluscs may also become toxic as cyanotoxins accumulate, moving up the food web [13,14,15,16]. Consequently, the dominance of cyanobacteria might cause considerable economic loss to fisheries and water supply companies, and decrease the recreational value of the water body and the market value of lakefront estates [10,17,18]. Rapidly changing environmental conditions may favor cyanobacterial growth over other species, with subsequent consequences such as habitat loss (e.g., hypoxic zones), disruption in energy flow along the food web, and decrease in biodiversity and ecosystem services [19,20,21]. 

Generally, cyanobacterial occurrence and dominance are significantly associated with three environmental variables and their interactions, nutrients, light and temperature, as reviewed in [22]. A high temperature stimulates cyanobacterial growth, with many species reaching their maximum growth rate above 25 °C [11,23]. As the nutrient dynamics in lakes is strongly affected by the external loading of phosphorus (P) and nitrogen (N) [24], the abundance and community composition of cyanobacteria are mainly linked to these two environmental factors [25]. Traditionally, high P concentration is considered as the main risk factor for cyanobacterial blooms [26]. In shallow lakes, the internal load is also considered as an important source of dissolved P that becomes directly available for phytoplankton growth [27,28]. Several studies have also demonstrated the significance of N on promoting cyanobacterial blooms. While there has been a debate about whether P or N control algal growth, the importance of the dual control of nutrients is widely recognized in water quality management [29,30,31]. Because the geographical distribution and both the magnitude and frequency of cyanobacterial blooms is increasing [32], it is urgent to accurately monitor potentially toxic cyanobacteria and improve our understanding of the environmental factors that promote bloom formation.

Conventional monitoring methods rely on inverted microscopy [33] and spectrophotometric measurements of chlorophyll-a (chl-a) to quantify phytoplankton, including cyanobacteria. Both of these methods are insufficient to detect toxin-producing cyanobacteria [34] because toxic and non-toxic strains within the same species are often morphologically identical and might co-exist within the same sample [13]. Traditionally, cyanobacterial biomass has been used as an indicator of potential toxin presence, disregarding the fact that toxin concentrations can rapidly increase even in a lower cyanobacterial biomass [34]. Measurements of chl-a and some other pigments provide a rapid estimation of total algal biomass (particularly, combined with remote sensing) [35], but, even if more group-specific photosynthetic pigments, such as zeaxanthin or phycocyanin, are used as a proxy for cyanobacterial biomass, the toxicity potential remains unknown.

To evaluate the risks to public health, data on potentially toxic cyanobacteria and cyanotoxins need to be provided as early as possible [11]. DNA based methods (e.g., polymerase chain reaction—PCR and quantitative PCR) comprise a valuable toolbox for both the detection and quantification of potentially toxic cyanobacteria [25,36,37,38]. However, the relationship between the cellular microcystin (MC) concentration and the copy number of *mcy* genes or MC producer’s biomass is not straightforward. The amount of microcystin per unit of biomass (toxin quota per biomass) depends on various factors such as genotype, nutrient availability and temperature. The general understanding about the amount of toxin per cell under different environmental conditions is still poorly understood due to the availability of only a limited number of in situ studies. As the toxin quota per cell directly reflects the safety of the waterbody, it is important to clarify how the copy number of *mcy* genes, MC concentration and environmental factors are related to this specific parameter. Quantitative PCR (qPCR) is a time- and cost-effective tool to evaluate the proportion of potential toxin-producing genotypes over the cyanobacterial population, as well as to investigate how environmental parameters determine toxicity potential at the spatio-temporal scale [25,34]. Consequently, qPCR can help us understand the mechanisms that trigger toxic blooms and can be used as an early warning tool for lake managers.

Lake Peipsi is the largest transboundary lake in Europe, located on the border of Estonia and Russia (Figure 1). This large (area 3555 km^2^), shallow (mean depth 7.1 m) and unstratified lowland water body consists of three basins (Figure 1, Table A1). The northernmost Lake Peipsi *sensu stricto* (*s.s.*) with a very simple shoreline is the deepest part of the lake. The southernmost basin is Lake Pihkva and these two lakes are connected by riverlike Lake Lämmijärv [39]. These three basins are all different in trophic state, hydrology and morphometry [40]. The bottom topography of Lake Lämmijärv is remarkably different to the other basins and also the water temperature in spring and winter tends to be warmer than in Peipsi *s.s*. and Lake Pihkva [39]. According to the OECD (1982) classification, Lake Peipsi *s.s.* is considered as eutrophic and Lämmijärv and Pihkva basins are considered as hypertrophic parts of the lake (Table 1, Figure 1). Rivers Velikaya and Emajõgi, the main inflows, carry the majority (>80%) of nutrients into the lake [41,42]. The outflow from the lake, River Narva, discharges into the Gulf of Finland. Lake Peipsi has been strongly influenced by eutrophication and natural fluctuations in water level and temperature [40]. Due to these processes, massive cyanobacterial blooms have been common for several decades [43,44,45].

In this study, samples from Lake Peipsi and its basins (Figure 1; Table A1) were analysed for the presence and abundance of potentially toxic *Microcystis*, *Dolichospermum* and *Planktothrix* using qPCR. The concentration and variants of microcystins were analysed using liquid chromatography-mass spectrometry (LC-MS/MS). Microscopic analysis of the samples was conducted to analyse the cyanobacterial community composition. Although several studies on the occurrence of cyanobacteria in Lake Peipsi have been published [41,44,46,47,48,49], only one of these addresses the issue of toxicity and toxin concentrations [49]. In general, there are significantly less data and fewer studies on cyanotoxins from Eastern Europe, with Estonia being presented with only one publication [50]. The current study is the first to combine molecular tools with traditional methods to determine the potential for cyanobacterial toxicity in this large and shallow north temperate lake. 

Here, we aim to (1) analyse cyanobacterial community composition in ecologically contrasting basins of Lake Peipsi and use molecular markers to identify and quantify the potential microcystin producers in the lake, (2) determine the relationship between toxin quota, the abundance of *mcyE* genes and MC concentrations, and (3) elucidate the environmental factors that promote toxic cyanobacterial blooms. Our first hypothesis is that the number of *mcyE* copies will follow an increase in toxin concentration. In our second hypothesis, we assume that in Lake Peipsi, *mcyE* gene copy number can be potentially used as a predictor of MC concentration. Additionally, we hypothesize that specific toxin variants are directly related to certain cyanobacterial genera.

## 2. Results

### 2.1. Environmental Variables 

During the study period, the basins of Lake Peipsi were characterized by different water quality parameters and general characteristics (Table 1). Across the lake basins, spatial gradients occurred in the trophic state. Total phosphorus (TP), soluble reactive phosphorus (SRP), total nitrogen (TN) and chl-a values increased from the northern basin towards the southern basins. 

### 2.2. The Composition of the Cyanobacterial Community Based on Microscopy

During the growing season (May–October) across the basins of Lake Peipsi, cyanobacteria and diatoms prevailed in the phytoplankton biomass, while chlorophytes and cryptomonads prevailed in abundance. During the period of 2010–2012, cyanobacterial biomass varied from 0–16.03 mg wet weight/L (mgWW/L) and dominated in the summer months (July–August) or early autumn (September). The main potentially toxic cyanobacteria in the lake were from N_2_-fixing heterocystous genera *Gloeotrichia*, *Dolichospermum*, *Aphanizomenon*, and non-heterocystous *Microcystis* and *Planktothrix*. The biomass of the genus *Microcystis* exceeded other genera manifolds (median 1.7 mgWW/L, maximum 14.5 mgWW/L in August), and was dominant among potentially toxic algae. *Planktothrix* attained a peak in September (maximum 2.2 mgWW/L) and *Dolichospermum* at the beginning of July, with maximum biomass of 2.4 mgWW/L (Figure 2). *Gloeotrichia* occurred sporadically in some regions of Peipsi *sensu stricto* (*s.s.*) and only a few times in Lämmijärv. *Aphanizomenon* was omnipresent in the lake, being more numerous in Peipsi *s.s.* Other potentially toxic genera appeared all over the lake with larger biomasses in southern basins, and lakes Lämmijärv and Pihkva [43]. Multivariate comparison between-groups principal component analysis (bgPCA) revealed a clear spatial distribution of cyanobacterial community composition in different basins of Lake Peipsi (permutation test, *p* < 0.01) (Figure 3). The cyanobacterial community composition in Peipsi *s.s.* varied considerably (permutation test, *p* < 0.01) from the communities in Lämmijärv and Pihkva. Additionally, the effect of the inflow from Emajõgi River was evident in station 38, which was significantly different (permutation test, *p* < 0.01) from the other areas (Figure 3).

### 2.3. Abundance of mcyE Genes

A simultaneous occurrence of the main MC producers was observed in all regions of the lake. In 80% of the samples, all three genera appeared concurrently. *Microcystis mcyE* genes were found in all of the samples (*N* = 141), *Dolichospermum mcyE* and *Planktothrix mcyE* were found in 95% and 83% of the samples, respectively. Compared to the other genera, *Microcystis mcyE* copy numbers were most abundant (Wilcoxon pairwise test, *p* < 0.01) over the entire growing season (median 1.89 × 10^5^ gene copy/mL, minimum 0 and maximum 2.6 × 10^7^ gene copy/mL, Figure 4). *Dolichospermum mcyE* and *Planktothrix mcyE* copy numbers were more comparable (median values 4.6 × 10^1^ and 2.3 × 10^2^ gene copy/mL; minimum 0 and 0, maximum values 4.05 × 10^4^ and 7.46 × 10^4^ gene copy/mL, respectively). Positive correlations were found between the biomass and the toxin-producing gene abundances for *Microcystis* (*r* = 0.6; *p* < 0.01), *Dolichospermum* (*r* = 0.31; *p* < 0.01) and *Planktothrix* (*r* = 0.62; *p* < 0.01).

The seasonal pattern of the values of toxic genotypes was consistent with the dynamics of the total MC concentrations (Figure 5a,b). The total microcystin concentration had a statistically significant positive correlation (r_p_ = 0.67; *p* < 0.01; *n* = 69) with the sum of *Microcystis*, *Dolichospermum* and *Planktothrix mcyE* gene copy numbers (Appendix A). The seasonal dynamics of the microcystin quota (particulate microcystin per unit of *mcyE* gene) was different from the seasonal dynamics of the MC concentration: a lower microcystin quota per *mcyE* gene occurred together with higher MC concentrations (Figure 5c) and vice versa. There was a strong negative correlation between MC quota per *mcyE* gene and *mcyE* gene copy number (*r* = −0.75; *p* < 0.01; *n* = 69). Logistic regression analysis demonstrated a correlation between MC-RR presence/absence and microcystin quota per *mcyE* gene (*z* = 2.36; *p* < 0.05); however, there was no statistically significant correlation between other MC variants and MC quota per *mcyE* gene. MC quota per *mcyE* gene was positively related to water temperature and pH (r_s_ = 0.46 and 0.31 respectively, *p* < 0.05; *n* = 69) negatively related to nitrate concentration (r_s_ = −0.30), yet no significant correlation was found between MC quota per *mcyE* gene and both TP and TN. The correlation with the toxin quota calculated per unit of chlorophyll-a resulted in similar relationships to these environmental parameters (r_s_ = 0.47 and 0.50, respectively, for temperature and pH, and r_s_ = −0.38 for nitrate; *p* < 0.05; *n* = 69).

According to the literature data [25,32], 36 species of cyanobacteria in Lake Peipsi are potentially toxic. A Mantel permutation test showed a significant relationship between the biomass of potentially toxic species and *mcyE* gene abundances (*r* = 0.43, *p* < 0.01, 999 permutations, *n* = 141). At the same time, there was no statistically significant correlation between the biomass of other non-toxin producing cyanobacteria and *mcyE* abundance (*r* < 0.01, *p* = 0.53, permutations *n* = 999, *n* = 141).

### 2.4. Microcystin- Concentrations and Variants

In this study, MCs were found in all samples analysed (*n* = 69). With a few exceptions, the microcystin concentration in the samples was relatively low, ranging from 0.001–10.9 µg/L, median 0.4 µg/L. The maximum concentration was measured in mid-July 2012 in Lämmijärv, when *Microcystis wesenbergii* (Komárek) Komárek ex Komárek dominated. Based on LC-MS/MS, a total of 14 MC variants were found and eight of them were identified (MC-RR; [D-Asp3]MC-RR; [Dha7]MC-RR; MC-LR; [D-Asp3]MC-LR; [Dha7]MC-LR; MC-YR; [Dha7]MC-YR). Microcystin-RR was the most abundant MC variant, found in 93% of samples, followed by MC-LR and its methylated variants in 92% of the samples. In 60% of the samples, all identified MCs co-existed. Both PCA analysis and linear fitting of environmental variables (Figure 6) demonstrate a clear pattern of the distribution of MC variants, gene copy numbers, and the biomass of potentially MC-producing species. During the entire growing season, all variants of MC were detected in samples from Lämmijärv, while all MC variants were detected in samples from Peipsi *s.s.* only during the late growing season. [D-Asp3]MC-RR was correlated with *Planktothrix agardhii* Gomont, Anagnostidis and Komarek in the southern basins (Lämmijärv and Pihkva) from July until the end of the sampling period. Other MC variants formed a close group with *M. wesenbergii*, *Microcystis aeruginosa* (Kützing) Kützing and *Dolichospermum flos-aquae* (Brébisson ex Bornet and Flahault) P. Wacklin et al. Concurrently, *Dolichospermum circinale* (Rabenh.) Wacklin *et al*., and *M. wesenbergii* were also highly related to *mcyE* gene copy numbers.

### 2.5. Environmental Variables that Favour Potentially Toxic Cyanobacteria Genotypes

A principal component analysis was performed using the cyanobacterial community composition as variables and linearly fitting the environmental variables, such as basic nutrients (TP, SRP, TN, NO_3_-, NO_2_-, NH_4_+) and temperature with a PCA ordination space. The analyses revealed a significant but mostly weak association with the distributions of environmental variables that favor cyanobacteria in Lämmijärv and Peipsi *s.s.* (Figure 7). In the early growing season, water temperature (r^2^ = 0.14; *p* < 0.05) and nitrate (r^2^ = 0.19; *p* < 0.01) were the main factors associated with cyanobacterial abundance in both lake basins. From August to October, in Peipsi *s.s.*, soluble reactive phosphorus (SRP) (r^2^ = 0.14; *p* < 0.05), and in Lämmijärv total nitrogen (r^2^ = 0.28; *p* < 0.001) and total phosphorus (r^2^ = 0.69; *p* < 0.01) were the most important environmental descriptors related to the cyanobacterial community composition. Other environmental parameters (e.g., NH_4_^+^, NO_2_^−^ etc.) were analysed as well, but no significant correlations were found with the cyanobacterial community.

## 3. Discussion

Cyanobacterial occurrence and dominance are mainly associated with eutrophication processes [4]. One of the initial aims of this study was to identify differences in the cyanobacterial community composition along a trophic gradient and to determine which cyanobacteria genus is the main potential microcystin producer in the shallow eutrophic–hypertrophic Lake Peipsi. The current study revealed a clear spatial distribution of cyanobacterial community composition in ecologically contrasting basins of this large and shallow lake. The cyanobacterial community composition in the eutrophic part of the lake varied considerably from the communities in hypertrophic basins (Figure 3, Table 1, Table A1). This observation is in accordance with a previous study that demonstrated remarkable differences between Peipsi *s.s.*, Lämmijärv and Pihkva [44]. The most abundant species were *Microcystis viridis* (A. Braun) Lemm., *M. wesenbergii*, *Dolichospermum. circinale*, *D. crassum* (Lemm.) Wacklin *et al*., *D. lemmermannii* (Richter) Wacklin *et al*., *Planktothrix agardhii*, *Aphanizomenon flos-aquae* Ralfs and *Gloeotrichia echinulata* (J.S. Smith, P. Richter). A more detailed overview of species composition is provided in Appendix A. Those findings are in agreement with data from systematic monitoring of Lake Peipsi, which has been carried out for six decades. According to the existing data [43], cyanobacterial blooms in the northern part of Peipsi *s.s.* are mainly attributed to *G. echinulata*. At the same time, several potential microcystin producers from genus *Microcystis*, *Dolichospermum* and *Planktothrix* occur in the southern part of Peipsi *s.s.* and Lämmijärv. *A. flos-aquae* has occurred in the lake in all years studied until the late autumn [43,44,45,46].

Another aim of this study was to use molecular markers to identify and quantify the potential microcystin producers in the basins of Lake Peipsi. For that purpose, genus-specific qPCR was used. Because microcystins are considered a major threat to human and animal health worldwide [51], we primarily focused on the genetic markers of the three most common microcystin-producing genera in Lake Peipsi (*Microcystis, Dolichospermum* and *Planktothrix*). Molecular analysis of potential MC producers revealed a simultaneous occurrence in all regions of the lake and 80% of the samples, with all three genera appearing concurrently (Figure 4). During the productive season (May to Oct), *Microcystis mcyE* copy numbers were the most abundant, while the abundance of *Dolichospermum mcyE* and *Planktothrix mcyE* copies were considerably lower. This result is not surprising because species from *Microcystis* are highly adaptive for various environments [52], and thus are the most commonly found microcystin producers in all eutrophic waters worldwide [13,51]. Although it is clear that the genus *Microcystis* was the dominant potential MC producer in Lake Peipsi, other genera also had their maximum peak periods (Figure 4). The ecological preferences of the different genera might provide an explanation. *Dolichospermum* had its peak in July, when the average water temperature was higher (mean 21.2 °C) and the water column more prone for stratification. Stratification periods give a competitive advantage for cyanobacterial species with gas-vacuoles (e.g., *Dolichospermum and Aphanizomenon*), as, due to their ability to regulate their position in the water column, they can optimise their use of nutrients and light [7]. However, it should be noted that with the given data it is difficult to assess the relative importance of direct temperature effects compared to the indirect effects or general climatic differences between seasons. In August, *Microcystis* dominated, and in September *Planktothrix* reached their peak. While water temperature in August was comparable with the temperature in July (mean 20.6 °C), in September the average temperature was only 14.4 °C. Compared to other genera, *Planktothrix* thrives in cooler water temperatures and is well adapted to lower temperatures [53]. This can explain its biomass peak in Lake Peipsi during the autumn months. Due to its low-light tolerance [54], *P. agardhii* can proliferate throughout a well-mixed shallow water column in the southern basins of the lake (Table A1), where the Secchi depth is significantly lower. 

In late summer, the abundance of cyanobacteria was primarily associated with the concentration of SRP (Figure 7) in Lake Peipsi *s.s.*, suggesting that the dominance of cyanobacteria and biomass of the major microcystin producer is mainly controlled by P dynamics. Similarly, the recent studies in Lake Peipsi and other shallow eutrophic lakes of the north temperate region have shown that internal P loading provides considerable amounts of bioavailable P to the water column, which contributes to the growth of cyanobacteria in summer [48,55,56,57,58,59]. Under turbid conditions and warm water temperatures, cyanobacteria gain an advantage over eukaryotic phytoplankton groups, as they can control their buoyancy to maximise the light use, maintain growth rates in warmer temperatures, fix atmospheric nitrogen, and therefore take advantage of the use that is provided by internal loading during N-limited periods [28,60]. Furthermore, several species of cyanobacteria are able to uptake and store bioavailable phosphorus, and thus the populations can sustain themselves on internal P storage [52]. Changes in the factors that regulate cyanobacteria abundance between early and late times of the growing season reported in the current study are most likely related to the changes in the relative importance of the sources of nutrient supply, as they are closely coupled to the seasonality of nutrient dynamics in Lake Peipsi [61,62]. Moreover, different cyanobacteria species may assimilate nutrients at different rates. This is supported by another of our findings: the clear spatial distribution of cyanobacterial community composition (Figure 3) in the basins of Lake Peipsi that we studied are characterized by the different trophy. The finding that NO_3_^−^ (Figure 7) was the main driver shaping the cyanobacterial community composition at the beginning of the growing season in all basins of Lake Peipsi, and TN, together with TP, are influential factors during the late growing season in Lake Lämmijärv, may imply that N also has to be considered in lake water quality management aimed at reducing cyanobacteria.

### 3.1. MC Concentration Versus MC Variants

During the study period, MC concentrations measured from lake water samples were in a range (median 0.4 µg/L) comparable to other various large lakes such as Taihu [63,64], Chaohu [65], Green Bay of Lake Michigan [66] and Erie [67,68]. In a study where 143 lakes in New Zealand were investigated, the authors also reported rather low MC concentrations (<1 µg/L) in the majority of lakes [69]. Comparable MC concentrations were also found in another large and shallow Estonian lake, Võrtsjärv [70]. Our toxin concentrations, measured in Lake Peipsi, were comparable with MC concentrations from the year 2003, reported in a study by Tanner et al. [49]. Tanner and others [49] demonstrated that even when the MC concentration in the open water column is relatively low, the MC concentration was extremely high in the inshore areas where biomass may accumulate and most human and animal activity occurs (33 to 54 times higher compared to the open water). Inshore samples were not analysed during the current study because MC concentrations were not measured, however, *mcyE* copy numbers in inshore waters were extremely high, reaching 57 million gene copies per mL [70]. Species from *Microcystis* and *Dolichospermum* are able to form surface scums and, under favourable environmental conditions (e.g., abundant sunlight, warm temperature and still water-column), the density of potentially toxic cells can rapidly increase within a few hours [1]. If the wind sweeps these scums to the shore, it can present a very high risk for the people using the waterbody recreationally. In 2002, the MC concentration measured in the shoreline scum of Lake Peipsi was 2183 µg/L, even when the MC concentration in the open water was rather low [49]. Therefore, we can conclude that even moderate concentrations of microcystins in the open water area of the lake can pose a high risk for bathers if, under the right conditions, surface scums form and concentrate in shoreline areas.

In Lake Peipsi, a total of 14 MC variants were found and eight of them were identified. The most abundant MC variants were MC-RR, found in 93% of the samples, and MC-LR with its variants, found in 92% of the samples. MC-RR, together with MC-LR and MC-YR and their variants are the most commonly reported microcystins [51,71] and MC-LR is quite often mentioned as the most frequent MC.

More variants and a higher concentration of microcystins are often found in more eutrophic waters [66]. This is in accordance with our findings, showing that all analysed variants of MCs were detected in more eutrophic Lake Lämmijärv during the entire growing season, while in Lake Peipsi *s.s.* they were detected only during the late growing season. In the hypertrophic part of Lake Peipsi, *P. agardhii* was only significantly related with [D-Asp3]MC-RR (Figure 6) and the abundance of *Planktothrix mcyE* genes showed a significant correlation to this MC variant. This finding is in accordance with the study [72], where Sivonen and others found that *Planktothrix* isolates from Finnish lakes were able to produce only one of two types of microcystins ([D-Asp3]MC-RR or [Dha7]MC-RR), and not other MC variants. In Lake Peipsi, other MC variants formed a group with *M. wesenbergii*, *M. aeruginosa* in the southern parts of the lake and *D. flos-aquae* in Peipsi *s.s.* Thus, our third hypothesis that specific toxin variants are directly related to certain cyanobacterial genera was mainly supported. In Lake Peipsi, the presence of MC-RR was associated with MC quota per *mcyE* gene, while other MC variants did not show any significant impact. One possible explanation for this might be that microcystin-RR was also the most abundant MC variant found in the samples. 

The microcystin concentration displayed a statistically significant positive correlation with the sum of *Microcystis, Dolichospermum* and *Planktothrix mcyE* gene copy numbers (Appendix A). These results are in accordance with our first hypothesis. This demonstrates that *mcyE* gene abundance could be used to estimate toxin production. A review from Pacheco and others [34] regarding the use of qPCR to assess the toxicity of cyanobacterial blooms showed that 22 studies out of 33 (years 2003–2015) reported a persistent positive correlation between *mcy* gene copies and MC concentrations. In 80% of the studies that adopted *mcyE* gene detection, positive correlations were found [34]. Additionally, under the framework of the European Multi Lake Survey (EMLS) [73,74], where lakes across Europe were sampled once in a snapshot approach in 2015, a strong significant correlation between the abundance of *mcyE* gene and MC concentrations was found in 200 lakes [75]. In the current study, the advantage of *mcyE* gene abundance for the prediction of MC production was also confirmed by the analysis of the relationship between microscopy counts of cyanobacterial species that are known to produce toxins, MC concentrations and *mcyE* gene abundance. Still, it should be considered that, even though the correlation of the MC concentration and *mcyE* gene abundance is very strong, the number of gene copies merely reveals the potential to produce the toxin, and does not indicate if the genes of interest are actively expressed and toxins are produced [63,76]. Despite the positive detection of potentially toxic genotypes of cyanobacteria, mutations can inactivate the genes involved in the biosynthesis of toxins, and thus hinder toxin production [77,78]. Therefore, to confirm the presence and estimate the concentration of cyanotoxins in the water, chemical analytical methods, such as LC-MS/MS or HPLC, are still required [79]. In order to elucidate the processes underlying toxin dynamics in more detail in this freshwater system, further exploration focusing on measuring the expression of toxin genes along with toxin concentration and other lake parameters would be necessary. 

### 3.2. Toxin Quota per McyE Gene

The concentration of toxins in the water is related to the abundance of toxin-producing species and the amount of toxin per cell. [80]. Generally, the toxin quota is described as the amount per unit of either biomass or chl-a [73,80]. In the current study, the microcystin quota was calculated as the microcystin concentration per unit of *mycE* gene, and we used this to elucidate the direct relationship between the abundance of toxin genes and MC concentration. Even though a significant positive connection between MC concentration and *mcyE* genes was found in Lake Peipsi, this study revealed that the dynamics of the MC quota per *mcyE* gene and MC concentration in the water was not concurrent (Figure 5). A lower microcystin quota per *mcyE* gene occurred together with higher MC concentrations and vice versa. A significant negative correlation was also found between MC quota per *mcyE* gene and the abundance of *mcyE* genes. Therefore, we found that our results only partly supported our second hypothesis, which states that *mcyE* copy number could be used as a direct predictor of MC concentration in the lake. Although there appears to be a correlation between *mcyE* gene numbers and microcystins, the variation in the cellular quota of microcystins may lead to under- or overestimation of the risk when merely based on *mcyE* gene numbers. This means that, in the situation where low numbers of *mcyE* genes correspond to a higher toxin quota per *mcyE* gene, the public health concern is higher, as it is subjected to a higher toxicity potential. If a low amount of toxic cells in the water can produce very high toxin concentrations in the water after cell lysis, then larger gene copies could rapidly reach extreme MC concentrations. This is in accordance with earlier observations where the MC quota is calculated with biomass or chl-a [73,80]. To estimate the risk for the water consumers, it is important to understand the reason behind the higher toxin quota when the biomass and *mcyE* gene copy numbers are rather low [80]. In response to environmental conditions, the toxin quota per cell in toxin-producing species can vary largely [13]. In our study, the MC quota per *mcyE* gene was related positively to water temperature and pH, and negatively to the concentration of nitrate, but no significant correlation was found with total nitrogen or total phosphorus. This finding broadly supports the work of other studies in this area that link the dynamics of toxin quota per biomass with water temperature. A similar conclusion was also reported by Wood and others [81] in a shallow eutrophic lake in New Zealand, where MC quotas per biomass responded positively to surface water temperature. In another large-scale study, where 137 European lakes were analysed, the authors also report no direct impact of TP and TN on the toxin quota per chl-a, but found that water temperature is an important control factor [73]. Several other studies have shown the importance of water temperature as the regulatory factor of cyanobacterial biomass [23,31,82,83] even on the global scale [84]. While nutrients play an important role through supporting the cyanobacterial community and biomass, the temperature seems to be to discriminative when other conditions are rather equal. These results demonstrate that, as global temperatures are expected to increase [85], in addition to an increase in the distribution, intensity, and duration of cyanobacterial blooms [23], the toxin concentration per cell will get higher. However, further in situ research is required to refine our understanding of the complex interaction between toxins, toxin quota per toxin gene and nutrients, as the findings about the effect of NO_3_^−^ on the toxin quota in general still seem contradictory. Our study suggests a negative correlation between nitrates and toxin quota per *mcyE* gene; one explanation for this inverse relationship can be the uptake of NO_3_^−^ by toxin-producing cyanobacteria. The study by Horst and others [80] demonstrates a positive relationship between these variables and [81] did not detect any significant relationships between toxin quota and environmental parameters (including nitrates) other than the water temperature. However, as the role of nutrients is more complex, the absence of the significant relationship with nutrient-related parameters does not mean that the distribution and concentration of toxins are not influenced by nutrients. In addition, the mentioned studies used toxin quota calculated per biomass of cyanobacteria. We assume that, in general, the biomass of cyanobacteria or the concentration of chl-a are less reliable predictors of toxin concentration and the predictive power can be increased by measuring the absolute abundance of toxin production genes. Expanding this knowledge in further in situ studies would substantially contribute to appropriate lake management and risk assessment of the toxic blooms.

To conclude, we demonstrated that, even though the number of *mcyE* gene copies increased together with toxin concentration, the variation in the cellular quota of microcystins may lead to under- or overestimation of the risk when merely based on *mcyE* gene copy numbers, and therefore *mcyE* copy number should not be used as a single measure to predict MC concentration in Lake Peipsi. Additionally, we showed that specific toxin variants were directly related to certain cyanobacterial genera. *P. agardhii* was significantly related with only [D-Asp3]MC-RR and other MC variants formed a close group with *M. wesenbergii*, *M. aeruginosa* in *D. flos-aquae*. Further, nitrate was the only nutrient-related variable connected to MC quota per *mcyE* gene. A strong positive correlation between water temperature and MC quota per *mcyE* gene suggests that the warming trends might lead to more harmful cyanobacterial blooms in temperate shallow lakes. 

## 4. Materials and Methods 

### 4.1. Study Site and Field Surveys

During the growing season (May–Oct), water samples from the Estonian part of Lake Peipsi *s.s.* and Lämmijärv were collected biweekly in 2011 and monthly in 2012. In addition, samples from the whole lake were collected from 15 sampling stations in August during the Estonian–Russian joint sampling campaigns from the period 2010–2012. The coordinates of the studied sampling points are shown in Appendix A. One hundred and forty-one depth-integrated water samples (depth range: surface to 0.5 m to the sediment) from 6–15 locations (Figure 1) were analysed.

Depth-integrated water was collected using a two-liter Van Dorn sampler at one-meter intervals and mixed in the collection bucket on the board of the research vessel. Integrated samples throughout the whole water column were collected due to the presence of buoyant cyanobacterial species. Subsamples for phytoplankton community composition analysis, molecular analyses of *mcyE* gene abundance, and toxin analysis were collected onboard. Subsamples were stored in an onboard refrigerator and transported to the laboratory in the coolers for further processing. For DNA extraction, 100–2000 mL (depending on sampling point and time) of the depth-integrated water was filtered at a low vacuum (max. 0.2 bar) through 5 µm pore size Whatman Cyclopore Polycarbonate filters. For toxin analyses, 150–1200 mL of the water was filtered through Binder-Free Glass Microfiber Grade GF/C (pore size 1.2 µm, GE Healthcare, UK) filter. Until further analysis, filters were stored at −80 °C.

Water chemistry analyses were performed as a part of the state monitoring programme by Estonian Environmental Research Centre following international and Estonian quality standards (ISO and EVS-EN ISO).

### 4.2. Microscopic Analysis

Samples for phytoplankton community analyses (*n* = 141) were preserved with Lugol’s (acidified iodine) solution and processed using the Utermöhl [33] method. Phytoplankton biomass was calculated from counts of cells using a Nikon Eclipse Ti-S inverted microscope at ×200 and ×400 magnification. Species were identified using classifications described in [86,87,88]. An aliquot of 3 mL was settled overnight. Biovolume of algal cells, colonies and/or filaments were calculated using assigned geometric shapes dimensions and converted to biomass assuming the specific density of 1 g/cm^3^ in accordance with [89].

### 4.3. Detection of Microcystins (MCs)

MCs were identified from 69 environmental samples by LC-MS according to their microcystin characteristic protonated molecular ions [M-H]^−^. The extracts were analysed (injection volume 5 μL) with an Agilent 1100 Series LC/MSD Trap System high-performance liquid chromatography (Agilent Technologies, PaloAlto, CA, USA.), which has an XCT Plus model ion trap as a mass detector. The ionization method used was electrospray ionization (ESI) in both positive and negative mode. The column used was Phenomenex Luna C8 (2) (150 by 2.0 mm, 5 μm) (Phenomenex, Torrance, CA, USA). Gradient was done with 0.1% formic acid in water (A) and 0.1% formic acid in 2-propanol (B) The gradient timetable was 20% B to 70% B over 37 min, after which the washing of the column was performed for in 100% B for 10 min and equilibrated in initial conditions for 12 min. The flow rate was 0.15 mL/min and the column temperature 40 °C. In ion source nebulizer gas (N2), pressure was 35 psi, desolvation gas flow rate 8 litres/min, and the desolvation temperature was 350 °C. The capillary voltage was set to 5000 V, the capillary exit offset was 300 V, the skimmer potential was 66 V, and the trap drive value was 73. Spectra were recorded 700–1500 m/z. The total microcystin concentration of the strain was approximated with a microcystin-LR standard (a gift from Z. Grzonka, Faculty of Chemistry, University of Gdansk, Poland) and microcystin-RR (Alexis, Farmingdale, NY, USA.) To avoid the underestimation of smaller concentration in environmental samples, standard curves were constructed with only the six most diluted standards. The identification of MCs was based on fragmentation patterns of the ions in MS2, ion masses and their retention times.

### 4.4. Cultivation of the Strains Used as External Standards

Microcystin-producing strains *Microcystis* sp. 205, *Dolichospermum* sp. 315 and *Planktothrix* sp. 49 were grown in HAMBI/UHCC Culture Collection, University of Helsinki in a Z8 medium under the continuous light at 20 ± 2. These strains were used in the preparation of standard curves for qPCR.

### 4.5. DNA Extraction

For DNA extraction, 40 mL of standard cultures were concentrated by centrifugation (5 min at 7000× *g*, at 4 °C) and DNA extracted immediately after that using E.Z.N.A.™ SP Plant DNA Kit (Omega Bio-Tek, Norcross, GA, USA) according to the manufacturer’s instructions. In addition to the protocol, the mechanical disruption of the cells with acid-washed glass beads (710–1180 µm; Sigma-Aldrich Co, St. Louis, MO, USA) and FastPrep® FP 120 bead-beater (MP Biomedicals, LLC, Irvine, CA, USA) was used. DNA from environmental samples was extracted using the DNeasy PowerWater Kit (Qiagen Inc., Germantown, MD, USA) according to the manufacturer’s instructions. The quality and quantity of extracted DNA were controlled with NanoDrop 2000 UV-Vis spectrophotometer (Thermo Fisher Scientific Inc., Waltham, MA, USA).

### 4.6. Detection and Quantification of mcyE Genes in Environmental Samples

To exclude false-negative outcome caused by the possible presence of PCR inhibitors in environmental samples, cyanobacterium-specific 16S rRNA PCR [90] was performed. PCR conditions are shown in Table A2. In order to identify the dominant potential microcystin-producing genera in environmental samples, genus-specific qPCRs were carried out. In order to detect *Planktothrix mcyE* genes in water samples, a new *Planktothrix*-specific primer pair and hydrolysis probe was designed (Appendix A). PCR conditions were optimized, and specificity (Appendix A) and sensitivity (Appendix A) experiments were performed as described before in [76].

The external standards of MC-producing strains were used to quantify *mcyE* gene copy numbers. Standard dilutions of genomic DNA of the standards were prepared as described before [91]. To quantify *Microsystis mcyE* gene copies from lake samples, the standard dilution contained 10^6^, 10^5^, 10^4^, 10^3^, 10^2^, 10^1^ copies of *mcyE* genes. To determine *mcyE* gene copies from *Dolichospermum* and *Planktothrix*, standard dilutions contained 10^5^, 2 × 10^4^, 4 × 10^3^, 8 × 10^2^, 1.6 × 10^2^, 3.2 × 10^1^ and 10^1^ copies of *mcyE* genes.

The reaction mixture in total volume of 20 µL included 5 µL of standard or environmental DNA, 1 × HOT FIREPol® Probe qPCR Mix Plus, with ROX (Solis BioDyne, Tartu, Estonia), 300 nM of both primers (Metabion international AG, Planegg, Germany) (Table A2) and 200 nM of TaqMan® probe (with an exception for *Planktothrix*, where 300 nM of TaqMan® probe was used). Environmental DNA samples were diluted 1:50 (to detect *Microcystis*) or 1:10 (to detect *Dolichospermum* and *Planktothrix*), MQ water was used as a negative control, and all the reactions were performed in three replicates. Amplifications were performed on an ABI 7500 Fast Real-Time PCR system (Thermo Fisher Scientific Inc, Waltham, MA, USA) using the following protocol: 95 °C for 12 min for initial denaturation, 40 cycles of 95 °C for 15 s and 62 °C (*Microcystis mcyE* and *Dolichospermum mcyE*) or 60 °C (*Planktothrix mcyE*) for 1 min. Results were analysed using 7500 Software version 2.0.5. 

The microcystin toxin quota was calculated by dividing particulate microcystin concentration (μg/mL) by *mcyE* gene copies/mL.

### 4.7. Statistical Analyses

All statistical analyses were performed with the R package and its extensions [92] and STATISTICA 13 (TIBCO Software Inc., PaloAlto, CA, USA). To analyse cyanobacterial community composition and to compare the communities from different lake basins (sampling stations), we used a between-groups principal component analysis (bgPCA, R ade4) on the abundance data. The statistical significance (*p* < 0.05) of bgPCA was tested by a Monte Carlo permutation test (1000 replicates; [93].

Wilcoxon pairwise test was used to analyse the differences between gene abundance of *Microcystis mcyE*, *Dolichospermum mcyE* and *Planktothrix mcyE*. Pearson correlation (r_p_) analysis on log-transformed data was used to test associations between *Microcystis*, *Dolichospermum* and *Planktothrix* specific *mcyE* gene abundance and total microcystin concentrations in the samples. In case the data lacked normal distribution even after logarithmic transformation, the Spearman rank-order correlation (r_s_) coefficient was used. 

Mantel test was performed to describe the relationship between matrices of microscopy counting data including cyanobacterial species potentially able to produce toxins and MC concentration. For this, the species list of cyanobacteria was compiled according to the historical records (provided by R. Laugaste) of phytoplankton species in Lake Peipsi. The potential ability to produce toxins was added from the list of toxin-producing cyanobacteria published by [1,25,32].

Multivariate comparisons of various *mcyE* gene abundances and presence/absence of MC variants were analysed using the subset of existing observations (*n* = 69) in PCA and linear fitting of independent variables (envfit function in R vegan). A similar analysis was used to determine the relationship between the abundance of cyanobacteria and environmental physico-chemical variables. The significance (*p* < 0.05) of these linear fittings was obtained by permutation test (1000 replicates).

The microcystin quota was calculated by dividing MC concentration by *mycE* gene variants abundance (copy number/mL). Thereafter, the association between microcystin quota per *mcyE* gene and MC variant presence/absence was analysed by logistic linear modelling (glm function with binomial family in R base).

## Figures and Tables

**Figure 1 toxins-12-00211-f001:**
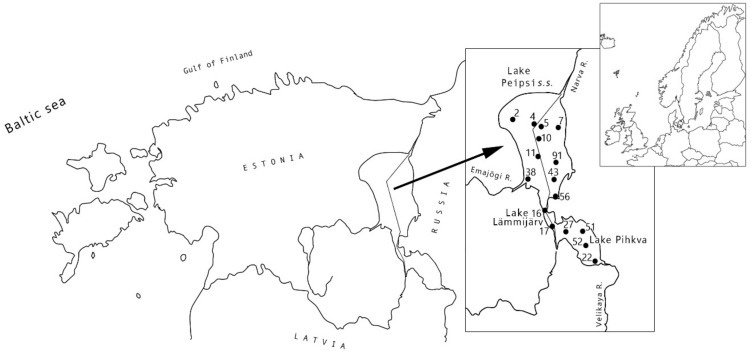
Location of Lake Peipsi (Estonia/Russia) and the sampling stations under study. Samples from Lake Pihkva (Russia) are collected in August only.

**Figure 2 toxins-12-00211-f002:**
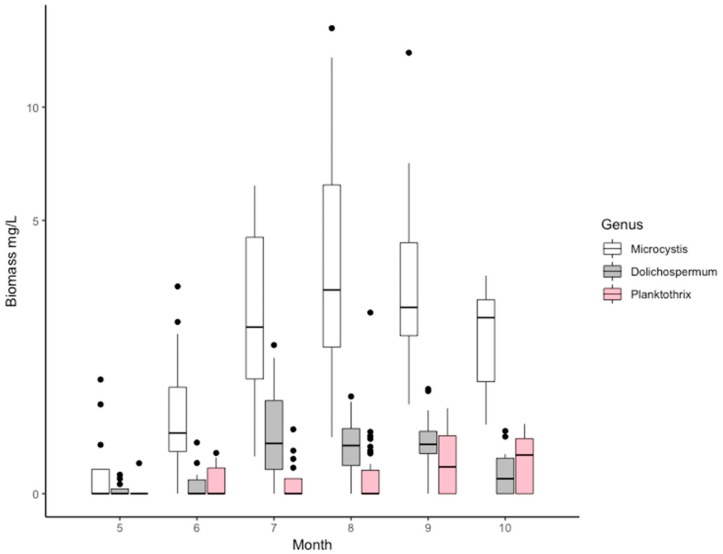
Temporal variation in *Microcystis*, *Dolichospermum* and *Planktothrix* biomass (mgWW/L). Boxplots denote median biomass values across the basins of Lake Peipsi and error bars represent spatial variation across the sampling stations. The y-axis is plotted as square root scale, values of biomass remain as original.

**Figure 3 toxins-12-00211-f003:**
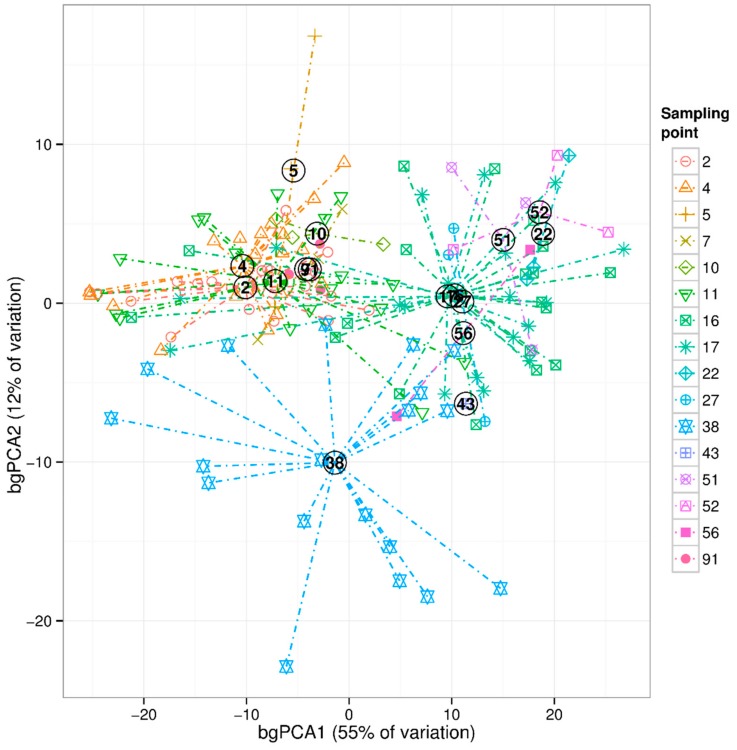
Cyanobacterial community composition in different basins (sampling stations) of Lake Peipsi. Sampling points 2, 4, 5, 7, 10, 11, 38, 43, 56 and 91 are located in Lake Peipsi *s.s*.; sampling points 16 and 17 in Lake Lämmijärv, and 22, 27, 51, 52 in Lake Pihkva.

**Figure 4 toxins-12-00211-f004:**
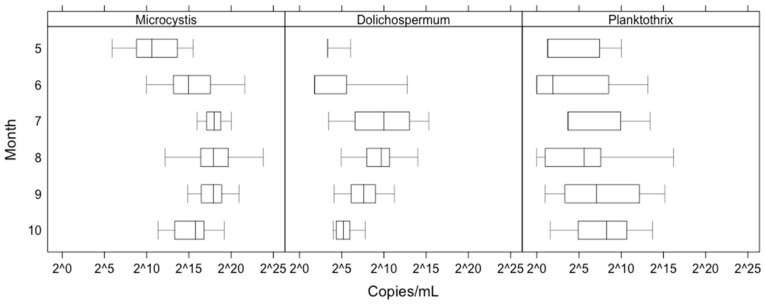
Temporal variation in *Microcystis*, *Dolichospermum* and *Planktothrix mcyE* gene copy number (*mcyE* gene/mL). Boxplots denote median *mcyE* copy numbers across the basins of Lake Peipsi. Error bars represent spatial variation across the sampling stations. On all three panels, the y-axis is plotted log_2_ scale and the values of *McyE* copies remain original.

**Figure 5 toxins-12-00211-f005:**
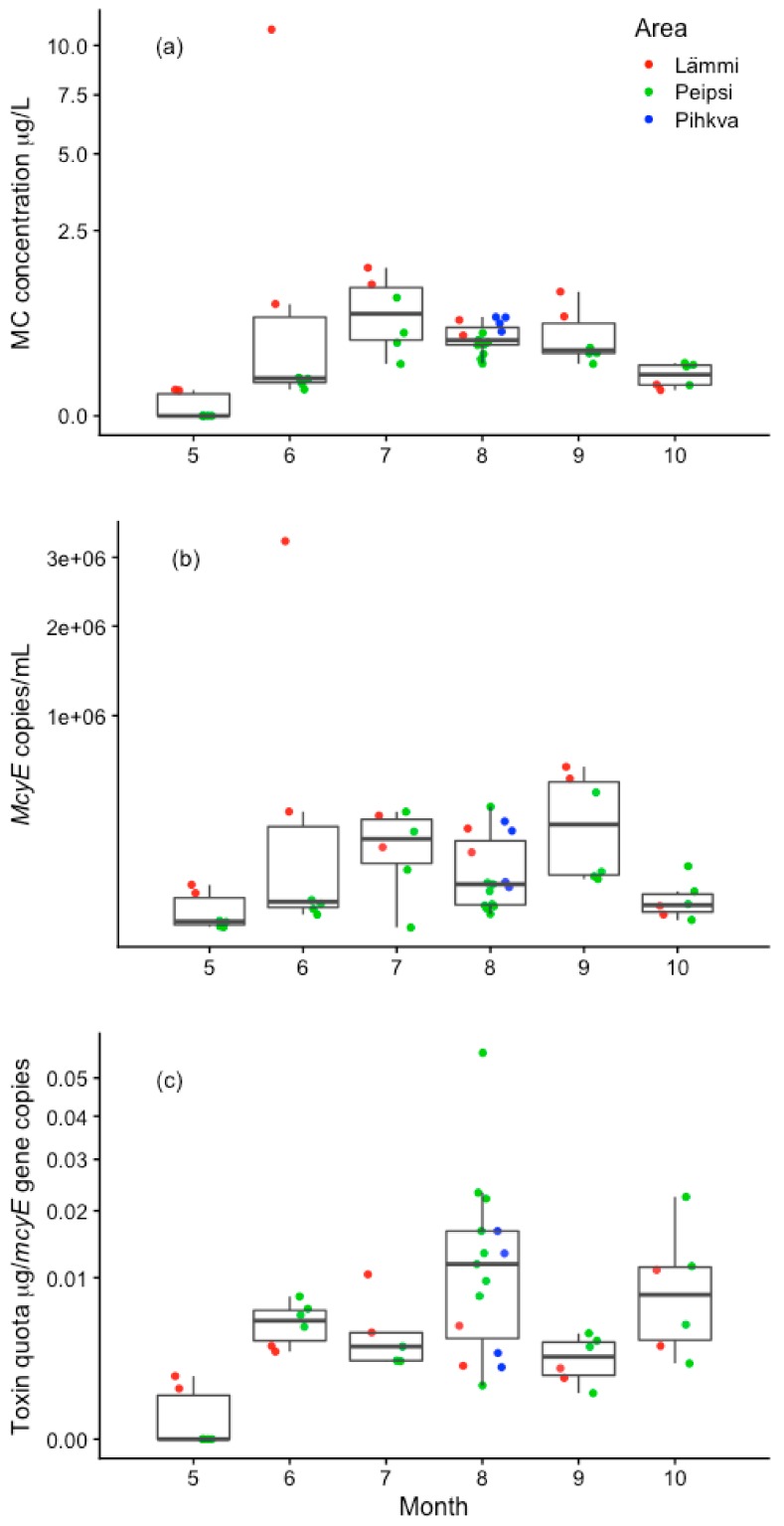
Temporal variation in total cell-bound microcystin (MC) concentration (**a**), the abundance of total *mcyE* genes (**b**) and toxin quota per *mcyE* gene—cell-bound MC per unit of *mcyE* gene (**c**) in the year 2012. Boxplots denote the median values of all basins and error bars represent spatial variation across all sampling stations. Points represent measurements in a specific lake basin. The y-axis is plotted as square root scale (MC concentration, *McyE* copies and Toxin quota per *mcyE* gene values remain original).

**Figure 6 toxins-12-00211-f006:**
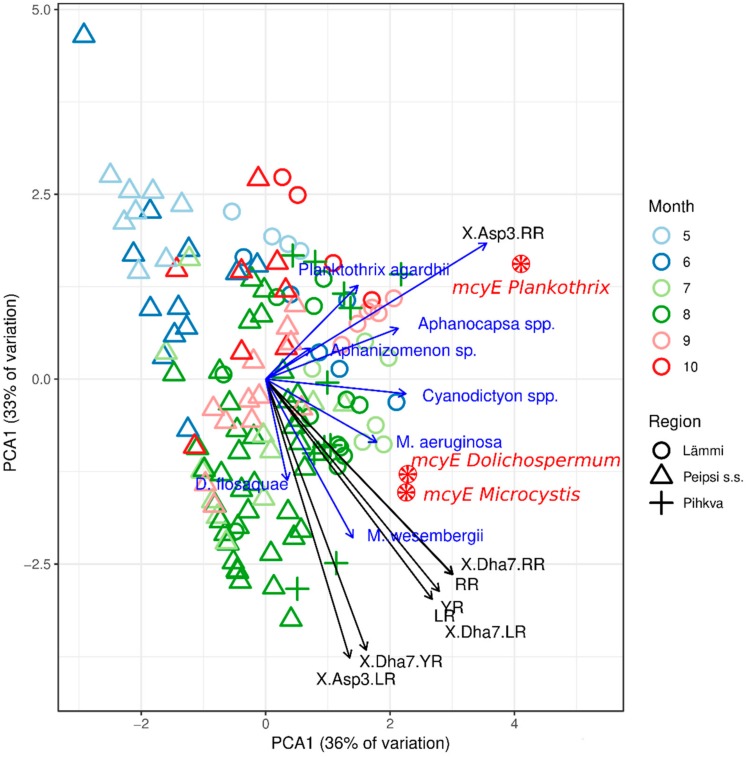
Multivariate comparisons of various *mcyE* gene abundances, cyanobacterial community and the presence/absence of MC variants. Significance (*p* < 0.05) of these linear fittings was obtained by a permutation test (1000 replicates). The length and direction of vectors indicate the strength and direction of the relationship.

**Figure 7 toxins-12-00211-f007:**
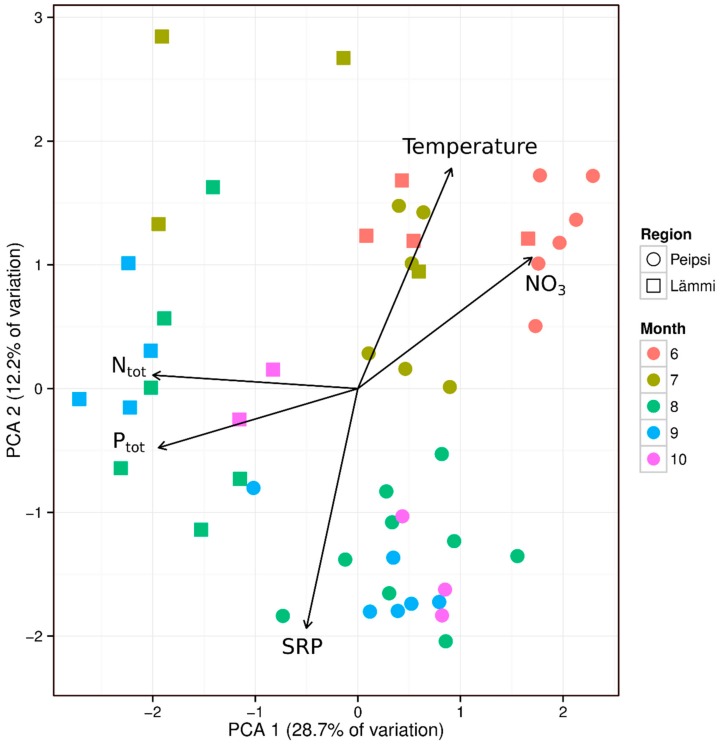
Multivariate comparisons of the abundance of cyanobacteria and environmental physico-chemical variables. Significance (*p* < 0.05) of these linear fittings was obtained by permutation test (1000 replicates). The length and direction of vectors indicate the strength and direction of the relationship.

**Table 1 toxins-12-00211-t001:** Water quality characteristics for three basins of Lake Peipsi (Lake Peipsi *sensu stricto*, Lake Lämmijärv, Lake Pihkva.

Characteristic	Peipsi *s.s.* *		Lämmijärv *		Pihkva **	
	Mean	Range	Mean	Range	Mean	Range
Number of Samples	91		38		12	
Area, km^2^	2611		236		708	
Mean depth, m	8.3		2.5		3.8	
Max depth, m	12.9		15.3		5.3	
Volume, km^3^	21.79		0.6		2.68	
TP, mg/m^3^	41	15–70	75	36–110	116	88–170
SRP, mg/m^3^	12	2–49	13	3–25	28	13–79
TN, mg/m^3^	701	460–1500	1001	410–1500	1147	950–1400
NO_3_^−^, mg/m^3^	91	15–930	115	30–820	91	30–220
NO_2_^−^, mg/m^3^	2	2–9	3	2–15	3	2–5
NH_4_^+^, mg/m^3^	28	10–162	25	10–120	24	10–58
chl-a, mg/m^3^	23.3	6.9–52.4	49.1	20.5–79	61.2	41.4–78.3
pH	8.5	8–8.9	8.6	8.3–9	8.9	8.4–9.2
Water temp, °C	18.2	5–23.9	17.9	10.3–24.7	22	19.7–25.6
Secchi depth, m	1.64	0.9–3.5	0.87	0.6–1.3	0.67	0.4–0.9
OECD classification	Eutrophic		Eutrophic/hypertrophic		Hypertrophic	

* mean values for growing season (2011–2012); ** mean values for August (2010–2012).

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
