# Peer review of "Using Microcystin Gene Copies to Determine Potentially-Toxic Blooms, Example from a Shallow Eutrophic Lake Peipsi"

_toxins, 2020, doi:10.3390/toxins12040211_

Round 1
Reviewer 1 Report
General comments
The manuscript is well written, clear and of interest for the area of cyanobacteria and cyanotoxins. The objectives are (1) use molecular markers to identify and quantify the potential microcystin producers in the ecologically contrasting freshwater basins of Lake Peipsi, (2) determine the relationship between toxin quota, abundance of mcyE genes and MC concentrations and (3) elucidate the environmental factors that promote toxic cyanobacterial blooms. The author’s hypothezed that the number of mcyE copies will follow an increase in cyanobacterial biomass and can potentially be used as a predictor of MC concentration in Lake Peipsi.
Most of the studies that address the abundance of toxic populations through the quantification of toxic genotypes by qPCR generally evaluate only one or two mcy genes, this leads to not detecting any possible genetic variety due to the bias of the primers. Could the authors have used other primers to amplify gene fragment for the synthesis of microcystins by qPCR? For example, mcyJ_F, mycJ_R, mycD_F, mycD_R, mycB_R, mycB_F, mycE_F, mycE_F described in bibliography (Kim et. Al., 2010, Kaebernick 2000, Hautala et. Al., 2013, Sipari et. Al., 2010. Please, clarify and / or justify the choice of a single primer.
There are several limitations to using gene quantification when compared to actual toxin measurements. It is possible for cyanobacteria to contain the target gene of interest, and therefore yield a positive result, but not produce toxin. This could be due to the absence of one or more of the required mcy genes via gene deletion, recombination, or transformation or due to gene disruption and inactivation by transposons or phage. Second, qPCR of DNA provides little information about the levels at which that gene is being expressed and whether, or how much of, the toxin is being produced.
Various literature works have concluded that quantification of toxin gene abundance is unlikely to provide data leading to a mechanistic or predictable ecological model of MC production in lakes.
It has been strongly suggested that future studies focus on measuring the expression of toxin genes, along with toxin concentrations, at higher resolution and include lake physics to elucidate the role of toxins in the freshwater systems. Please contemplate this suggestion in your manuscript.
Particular comments
Line 11-12 These sentence should be written more smoothly “Molecular tools e.g. qPCR can improve the understanding about mechanisms triggering the toxic blooms and can be used as an early warning tool for lake managers”
Figure 5 particulate MC concentration, please clarify what is particulate MC??. Toxin quota? Please include units in the quota and Fig. 5A is concentration the Microcystin variant, or total Micrcystin it is not clear in the figure and in the manuscript.
SRP in abstract section was not defined.
Table A1 Key characteristics of different basins of Lake Peipsi: You must report nitrate, nitrite and TN levels separately as well as SRP and TP not present in the table.
Author Response
Dear Reviewer,
Thank you for giving us the opportunity to respond to your questions and improve our manuscript. We found the comments and suggestions very constructive and helpful.
Dear Reviewer,
Thank you for giving us the opportunity to respond to your questions and improve our manuscript. We found the comments and suggestion very constructive and helpful. Please find below our detailed responses to the review.
General comments
The manuscript is well written, clear and of interest for the area of cyanobacteria and cyanotoxins. The objectives are (1) use molecular markers to identify and quantify the potential microcystin producers in the ecologically contrasting freshwater basins of Lake Peipsi, (2) determine the relationship between toxin quota, the abundance of mcyE genes and MC concentrations and (3) elucidate the environmental factors that promote toxic cyanobacterial blooms. The author’s hypothesized that the number of mcyE copies will follow an increase in cyanobacterial biomass and can potentially be used as a predictor of MC concentration in Lake Peipsi.
- Most of the studies that address the abundance of toxic populations through the quantification of toxic genotypes by qPCR generally evaluate only one or two mcy genes, this leads to not detecting any possible genetic variety due to the bias of the primers. Could the authors have used other primers to amplify gene fragment for the synthesis of microcystins by qPCR? For example, mcyJ_F, mycJ_R, mycD_F, mycD_R, mycB_R, mycB_F, mycE_F, mycE_F described in the bibliography (Kim et. Al., 2010, Kaebernick 2000, Hautala et. Al., 2013, Sipari et. Al., 2010. Please, clarify and/or justify the choice of a single primer.
Authors: Thank you for this observation as it allows us to clarify our work. In our study, the mcyE gene was chosen because it has been demonstrated that this gene plays a key role in microcystin production. Previous studies have shown that this specific region can reliably detect various microcystin producers in environmental samples. These qPCR primers used in our study were designed by Sipari et al., 2010 for Microcystis and Dolichospermum and by the authors of our study for Plantothrix (supplemental tables and files are provided with the manuscript).
- here are several limitations to using gene quantification when compared to actual toxin measurements. It is possible for cyanobacteria to contain the target gene of interest, and therefore yield a positive result, but not produce toxin. This could be due to the absence of one or more of the required mcy genes via gene deletion, recombination, or transformation or due to gene disruption and inactivation by transposons or phage. Second, qPCR of DNA provides little information about the levels at which that gene is being expressed and whether, or how much of, the toxin is being produced. Various literature works have concluded that quantification of toxin gene abundance is unlikely to provide data leading to a mechanistic or predictable ecological model of MC production in lakes. It has been strongly suggested that future studies focus on measuring the expression of toxin genes, along with toxin concentrations, at higher resolution and include lake physics to elucidate the role of toxins in the freshwater systems. Please contemplate this suggestion in your manuscript.
Authors: Thank you for your broad comment. Our results led to the same conclusion as reviewers proposed. We included a couple of sentences about the concerns and limitations about using gene quantification by qPCR and in further studies, it should be considered and rather the expression of toxin genes should be used. Lines 359 – 363: “Still, it should be considered that even though the correlation of the MC concentration and mcyE gene abundance is very strong, the number of gene copies merely reveals the potential to produce toxin, and does not indicate if the genes of interest are actively expressed and toxins are produced. Despite the positive detection of potentially toxic genotypes of cyanobacteria, mutations can inactivate the genes involved in the biosynthesis of toxins and thus hinder toxin production.”
Particular comments
- Line 11-12 These sentence should be written more smoothly “Molecular tools e.g. qPCR can improve the understanding about mechanisms triggering the toxic blooms and can be used as an early warning tool for lake managers”
Authors: The sentence was rephrased accordingly.
- Figure 5 particulate MC concentration, please clarify what is particulate MC??. Toxin quota? Please include units in the quota and Fig. 5A is concentration the Microcystin variant, or total Microcystin it is not clear in the figure and in the manuscript.
Authors: Particulate MC is cell-bound (intracellular) microcystin. In our study, only the intracellular toxin content from filtered samples was analysed. Toxin quota – in our study we considered toxin quota as particulate microcystin per unit of the mcyE gene. Microcystin toxin quota was calculated by dividing particulate microcystin concentration (μg/mL) by mcyE gene copies/mL. The unit was added to Figure 5: µg/mcyE gene copies.
We were not able to measure the concentration of MC variants, only presence/absence was determined. Therefore, the Fig. 5A reports only total MC concentration, not certain MC variant concentration. Corresponding marks were added to the text and figure caption (lines 176-177).
- SRP in the abstract section was not defined.
Authors: Abbreviation is defined in the abstract now.
- Table A1 Key characteristics of different basins of Lake Peipsi: You must report nitrate, nitrite and TN levels separately as well as SRP and TP not present in the table.
Authors: Mentioned characteristics were added to Table 1 in results and Table A1 in appendixes.
Reviewer 2 Report
Review of Toxins 72130
February 14, 2020
General comments:
This paper is way too long. It takes two pages for the author or authors to get to the purpose of their work. The statements for the aim of the research are confusing and, it turns out, they don’t address their hypotheses directly anywhere later in the manuscript.
Most disturbing to me as a reader is that they find results that are contrary to either their hypothesis or previous work and don’t comment on it. For example, in line 164 they state that the toxin quotas had different dynamics than the mcyE gene copy numbers That’s directly in contrast to their first hypothesis. They also find that the toxin quota doesn’t correlate to what they thought would be the major drivers, nitrate and total phosphorus. As a reader, I need to be presented with this information and then have it explained. If they have no coherent explanation, they shouldn’t publish the work.
Line by line comments and suggestions:
I’m providing some line-by-line comments because I have them, but I feel that the whole paper needs to be restructured.
8- I don't understand this phrase, do you mean high human demands?
10- triggers aren't the same as correlations to changes in macro nutrients or seasons
12- need rapid turn-around time for qPCR measurements - was this accomplished?
66 – You either need money, time or both. Taxonomic experts can be replaced by flow-through counters. You need experts to handle qPCR or mass-spec analyses.
77 - You didn't mention pigment studies in this paragraph. When you mention it in the next paragraph you left off the benefits that these can be combined with remote sensing, are fast and can be easily automated.
91 – you haven’t defined toxin quota yet, and since you’re using two different definitions in this paper, that’s crucial. You use the term both as toxins/biomass and toxins/copy-number.
112 - Can't even see Lake/Basin Lammjarv on this map, or the rivers.
123 - It's not really a peak if it lasts three months, that’s just seasonal growth.
132 - do you mean in the main basin of the lake. This is confusing because some maps call these three different lakes but you are referring to them as basins of one lake.
138 - Where is this river? Do you have any other current flow information for these lakes?
144 - It would help if this legend identified which sampling points are in which basins.
146 – page 5 - Five pages in before we see any data that relates to your hypotheses.
149 - number of samples for different factors differ across that paper, this is confusing There are 141 lake samples but only 69 MC measurement samples. How did you choose to limit these?
151 - Figure 4 is a log scale, there is no zero. Two things, figure axis is too small to see and it's essentially in base 2. But most importantly, are any of these significantly different? That's usually denoted with an asterix.
172 - I think this contradicts one of your statements in the introduction. Needs explanation.
172 - Do you mean that there was no significant correlation to TP, TN, or Chla? That’s your other hypothesis negated. Need to comment or explain.
181 - Figure 5 c - can't have zero on a log scale Y axis
184 - Looks like you only have 4 points in August for Lake Pihkva. You can’t really say much with only that number of samples.
186 - You don't have a goal or hypothesis relating to studying the variants. What is the purpose of this section. Is it methodological?
195 - to test hypothesis - you need to look at samples that might disprove your hypothesis.
211 - What is the cyanobacterial community variable, is it the sum of all the cyanobacteria by visual counts?
213 - clear pattern is overstating it - these correlation values seem very small
232 – I don’t I don't think you've shown anything about the basins in your lakes. It wasn't in your hypotheses either unless that’s what you meant about nutrient gradient.
249 - this seems like it belongs in the introduction, but the introduction is already too long
319 - needs a lead sentence that tells what this is about
341-344 - doesn't this contradict your beginning statements about rapid detection?
347 - This section is repetitive of previous statements, around line 160 that first described figure 5
376 - You only need this if you are going to make a predictive model and you have nitrate concentrations - I thought you were focusing on measuring the distribution of toxins
378 - Why was your study different?
Author Response
Dear Reviewer,
Thank you for giving us the opportunity to respond to your questions and improve our manuscript. We found the comments and suggestions very constructive and helpful.
Dear Reviewer,
Thank you for giving us the opportunity to respond to your questions and improve our manuscript. We found the comments and suggestion very constructive and helpful. Please find out responses below as point-to-point actions made to address your concerns
General comments:
This paper is way too long. It takes two pages for the author or authors to get to the purpose of their work. The statements for the aim of the research are confusing and, it turns out, they don’t address their hypotheses directly anywhere later in the manuscript.
Authors: We thank the reviewer for this useful comment that allowed us to create a more concise manuscript. More specifically, the Introduction was made more compact and the purpose of the study more clearly presented (lines 114 – 121). We critically re-evaluated our aims and hypotheses and tried to keep the focus on them in all sections of the manuscript. Aims one and two are addressed in the results section 2.3, where the abundance of mcyE genes in the whole lake is described. The first aim was indeed poorly addressed in the discussion section and we tried to improve it in the paragraph two (lines 262 – 270). Third aim is addressed in the results section 2.5 and in Discussion lines 285 – 305.
Most disturbing to me as a reader is that they find results that are contrary to either their hypothesis or previous work and don’t comment on it. For example, in line 164 they state that the toxin quotas had different dynamics than the mcyE gene copy numbers That’s directly in contrast to their first hypothesis. They also find that the toxin quota doesn’t correlate to what they thought would be the major drivers, nitrate and total phosphorus. As a reader, I need to be presented with this information and then have it explained. If they have no coherent explanation, they shouldn’t publish the work.
Authors: Thank you for your comment. We tried to address the contradictions with our initial hypotheses more precisely. We tried to rephrase the hypotheses in a clearer manner. Therefore, in this version of the manuscript, we formulated three hypotheses instead of the original two. Our results did confirm the first hypothesis. We added a corresponding remark to the Discussion (lines 345-347). However, our results did not support the second hypothesis, therefore, explanations were added in the Discussion section (lines 375-381). Our third hypothesis that specific toxin variants are directly related to certain cyanobacterial genera was mainly supported by our result. Explanations were added to the Discussion (lines 342 – 343). Additionally, we provide a clear-cut Conclusions section at the end of MS to assist the reader to get our message in a more concise way. Lines 413-421: To conclude, we demonstrated that eventhough generally the number of mcyE gene copies increased together with toxin concentration, mcyE copy number could not be used as a direct predictor of MC concentration in Lake Peipsi. Additionally, we showed that specific toxin variants were directly related to certain cyanobacterial genera. P. agardhii was significantly related with only [D-Asp3]MC-RR and other MC variants formed a close group with M. wesenbergii, M. aeruginosa in D. flos-aquae. Further, nitrate was the only nutrient related variable connected to MC quota per mcyE gene. Strong positive correlation between water temperature and MC quota per mcyE gene suggests that the warming trends might lead to more harmful cyanobacterial blooms in temperate shallow lakes.
.
Line by line comments and suggestions:
I’m providing some line-by-line comments because I have them, but I feel that the whole paper needs to be restructured.
8- I don't understand this phrase, do you mean high human demands?
Authors: Thank you for your comment. The confusing sentence was rephrased.
10- triggers aren't the same as correlations to changes in macro nutrients or seasons
Authors: Thank you for the comment. This sentence was rephrased.
12- need rapid turn-around time for qPCR measurements - was this accomplished?
Authors: yes, please see the next comment.
66 – You either need money, time or both. Taxonomic experts can be replaced by flow-through counters. You need experts to handle qPCR or mass-spec analyses.
Authors: Thank you for your comment. We agree that science does not yet have fully robotized methods while the methods we frequently use, require the intervention of experts. However, we argue here that some approaches can be more efficient than others. With regards time, the main advantage is the possibility to analyse several samples simultaneously. Depending on the machine used and the number of replicates, it is possible to analyse approximately 100 samples on one run with 384 qPCR platform
77 - You didn't mention pigment studies in this paragraph. When you mention it in the next paragraph you left off the benefits that these can be combined with remote sensing, are fast and can be easily automated.
Authors: Thank you for the comment, in order to shorten the introduction this paragraph was restructured and the pigment studies are now not addressed in detail in the manuscript. Short sentence “Measurements of chl-a and some other pigments provide a rapid estimation of total algal biomass (particularly, combined with remote sensing)” about the benefit to use pigments is added (lines 67-69).
91 – you haven’t defined toxin quota yet, and since you’re using two different definitions in this paper, that’s crucial. You use the term both as toxins/biomass and toxins/copy-number.
Authors: Thank you for the comment. In the MS we will now use the terms as follows: MC quota per biomass/Chl-a and MC quota per mcyE genes
112 - Can't even see Lake/Basin Lämmjarv on this map, or the rivers.
Authors: Figure 1 has been changed to improve the visualisation of the Lake/Basin Lämmijärv and the rivers. (Line 111)
123 - It's not really a peak if it lasts three months, that’s just seasonal growth.
Authors: Thank you for pointing out. This sentence was rephrased. (Line 137)
132 - do you mean in the main basin of the lake. This is confusing because some maps call these three different lakes but you are referring to them as basins of one lake.
Authors: Lake Peipsi sensu lato consists of three parts (the largest northern part – Lake Peipsi sensu stricto, the southern part – Lake Pihkva and narrow Lake Lämmijärv) (Jaani, 2001). Figure 1 has been changed and hopefully more understandable now. (Line 111)
Jaani, A. (2001). The location, size and general characterization of Lake Peipsi and its catchment area. T. Nõges (Toim), Lake Peipsi: Meteorology , Hydrology , Hydrochemistry (lk 10–17). Sulemees Publishers.
138 - Where is this river? Do you have any other current flow information for these lakes?
Authors: On the new map (Figure 1, line 111), river names were added.
River Emajõgi is the largest river discharging into Peipsi from the Estonian part of its watershed and River Velikaya from the Russian part of its watershed. These two rivers are carrying the majority of nutrients (>80%) into the lake (Loigu et al., 2008; Noges et al., 2010). There are many small inflows but these do not play an important role, therefore, they are not indicated on the map. The clarification about inflows was added to the Introduction on lines 96-97
Loigu, E., Leisk, Ü., Iital, A., & Pachel, K. (2008). Pollution load and water quality of the Lake Peipsi basin. Peipsi. (Eds J. Haberman, T. Timm & A. Raukas), 179–199.
Noges, T., Tuvikene, L., & Noges, P. (2010). Contemporary trends of temperature, nutrient loading, and water quality in large Lakes Peipsi and Vortsjärv, Estonia. Aquat. Ecosyst. Health Manag., 13(2), 143–153.
144 - It would help if this legend identified which sampling points are in which basins.
Authors: Thank you for pointing this out. Legend was modified accordingly. Basins and sampling points are now connected in the legend. Lines 157-159: Cyanobacterial community composition in different basins (sampling stations) of Lake Peipsi. Sampling points 2, 4, 5, 7, 10, 11, 38, 43, 56 and 91 are located in Lake Peipsi s.s.; sampling points 16 and 17 in Lake Lämmijärv and 22, 27, 51, 52 in Lake Pihkva.
146 – page 5 - Five pages in before we see any data that relates to your hypotheses.
Authors: Thank you for the comment. The introduction was shortened and restructured (lines 27 – 121). However, please pay attention that the hypotheses are at the end of the Introduction (118 – 121) and the Results start already on the line 124. Mentioned paragraph on line 160.
149 - number of samples for different factors differ across that paper, this is confusing There are 141 lake samples but only 69 MC measurement samples. How did you choose to limit these?
Authors: Thank you for the comment. For MC analyses, unfortunately, we were depending on the collaboration with the foreign laboratory and due to technical reasons we were able to collect samples for MC analyses after the study had already started. Therefore we have had fewer samples for MC analyses.
151 - Figure 4 is a log scale, there is no zero. Two things, figure axis is too small to see and it's essentially in base 2. But most importantly, are any of these significantly different? That's usually denoted with an asterisk.
Authors: Thank you for pointing out. The correction has been made on Figure 4 and in Figure 4 legend (lines 173 – 175). Clarification about X axis transformation has been added. Font size has been changed accordingly. As the lines 164 - 166 include the results of the Wilcoxon pairwise test, which concludes that Microcystis mcyE copy numbers are significantly different from other genera. Therefore, we feel that this fact is already presented in the manuscript and the purpose of the graph is mainly to illustrate the dynamics of mcyE gene copy numbers of different groups during the growing season.
172 - I think this contradicts one of your statements in the introduction. Needs explanation.
Authors: Explanations are not given in results part, please refer to Discussion lines 389-393
172 - Do you mean that there was no significant correlation to TP, TN, or Chla? That’s your other hypothesis negated. Need to comment or explain.
Authors: Thank you for the comment. We tried to address the contradictions with our initial hypotheses more precisely. We tried to rephrase the hypotheses in a clearer manner. Therefore, in this version of the manuscript, we formulated three hypotheses instead of the original two. Our results did confirm the first hypothesis. We added a corresponding remark to the Discussion (lines 345-347). However, our results did not support the second hypothesis, therefore, explanations were added in the Discussion section (lines 375-381). Our third hypothesis that specific toxin variants are directly related to certain cyanobacterial genera was mainly supported by our result. Explanations were added to the Discussion (lines 342 – 343). Additionally we provide a clear cut Conclusions section at the end of MS to assist the reader to get our message in a more concise way. Lines 413-421: To conclude, we demonstrated that even though, generally the number of mcyE gene copies increased together with toxin concentration, mcyE copy number could not be used as a direct predictor of MC concentration in Lake Peipsi. Additionally, we showed that specific toxin variants were directly related to certain cyanobacterial genera. P. agardhii was significantly related with only [D-Asp3]MC-RR and other MC variants formed a close group with M. wesenbergii, M. aeruginosa in D. flos-aquae. Further, nitrate was the only nutrient related variable connected to MC quota per mcyE gene. Strong positive correlation between water temperature and MC quota per mcyE gene suggests that the warming trends might lead to more harmful cyanobacterial blooms in temperate shallow lakes.
181 - Figure 5 c - can't have zero on a log scale Y-axis
Authors: Thank you for your comment. On y-axis scale square root transformation was applied. Clarifying remark was added to the legend of Figure 5. Lines 201 – 202: On y-axis scale square root transformation is applied (MC concentration, McyE copies and Toxin quota per mcyE gene values remain original).
184 - Looks like you only have 4 points in August for Lake Pihkva. You can’t really say much with only that number of samples.
Authors: Unfortunately, Lake Pihkva (Russia) was only sampled in August during the Estonian–Russian joint expeditions, other times the access to Russian territory was not possible. We do agree that this number of samples does not allow us to make fundamental conclusions about Lake Pihkva. However, we feel that it is still better to add this small set of samples to our study than not.
186 - You don't have a goal or hypothesis relating to studying the variants. What is the purpose of this section? Is it methodological?
Authors: Thank you for the comment, unfortunately, that MC variants were mistakenly left out from hypothesis. We rephrased our hypotheses accordingly. Lines 120 – 121: “In addition, we hypothesize that specific toxin variants are directly related to certain cyanobacterial genera.”
The purpose of studying toxin variants was to understand how mcyE genes and microscopy results are related to the MC variants.
195 - to test the hypothesis - you need to look at samples that might disprove your hypothesis.
Authors: Thank you for the comment. Unfortunately, the access to suggested samples has not been possible, due to financial and practical limitations to access the lake whole year around (ice cover period, ice thickness, need for hovercraft etc.). Also, as Lake Peipsi is transboundary lake we have access to the Russian side of the waterbody only during the Estonian-Russian joint expeditions. Therefore, we have used common approach using statistical tests whether we can reject or not the null hypothesis, e.g. evaluating ordination results by permutation test – observed data are compared to random sets of permuted data.
211 - What is the cyanobacterial community variable, is it the sum of all the cyanobacteria by visual counts?
Authors: Yes, it is Cyanobacterial Community Composition based on microscopy, as we do not have any other data.
213 - the clear pattern is overstating it - these correlation values seem very small
Authors: We agree that “clear pattern” is an overstatement and we modified these sentences accordingly. Lines 230 - 232 “The analyses revealed a significant but mostly weak association with the distributions of environmental variables that favour cyanobacteria in Lämmijärv and Peipsi s.s.”
232 – I don’t think you've shown anything about the basins in your lakes. It wasn't in your hypotheses either unless that’s what you meant about nutrient gradient.
Authors: We added the background information about the lake basins in the Introduction (Lines 87 - 100). The variability of water quality characteristics for three basins of the lake during the study period are shown in Table 1. Long-term characteristics are presented in Table A1.
235- Lines 235-240 very bored discussion
Authors: Thank you for pointing out! These sentences have been removed from Discussion.
249 - this seems like it belongs in the introduction, but the introduction is already too long
Authors: Thank you for the comment, the sentence was removed from this section.
319 - needs a lead sentence that tells what this is about
Authors: Lead sentence to the paragraph was added on lines 332-335.
341-344 - doesn't this contradict your beginning statements about rapid detection?
Authors: We do think that genes for potential toxicity are still rapidly detected using qPCR, another issue is how informative it is without gene expression and toxin analysis. Therefore, it is important to point out the shortcoming the method.
347 - This section is repetitive of previous statements, around line 160 that first described figure 5
Authors: Regrettably we do not fully understand the meaning of this comment, around lines 160 the results are presented without any statements.
376 - You only need this if you are going to make a predictive model and you have nitrate concentrations - I thought you were focusing on measuring the distribution of toxins
Authors: Thank you for the comment, we feel that in further in situ studies are required, to improve our understanding how nutrients and toxin concentration per cell are related in this specific large and eutrophic lake.
378 - Why was your study different?
Authors: We support that biomass of cyanobacteria or chl-a measurements in general are less reliable predictor for toxin concentration and our verdict is that the predictive power is increasing from measuring the absolute abundance of toxin production genes. We rewrote the ending of discussion accordingly. Lines 406 – 409: One possible explanation could be that studies mentioned used toxin quota calculated per biomass of cyanobacteria. We assumed that in general, the biomass of cyanobacteria or the concentration of chl-a are less reliable predictors of toxin concentration and the predictive power can be increased by measuring the absolute abundance of toxin production genes.
Reviewer 3 Report
Introduction
First, the title and the structure of the Introduction are not good. The authors give a strange literature review, discussing well-known facts, and often give not very correct and indirect references. It would be better if this lake, the fifth largest in Europe, were more clearly emphasized:
the name should have no parentheses, and the ecosystem of this important transboundary object should be described not in the penultimate paragraph but at the beginning of the article. Highlighting the significance of the study of cyanobacteria in freshwater ecosystems does not require a detailed
and partly unnecessary literature review on toxic blooms. Many references are incorrect. For example, Chorus and Bartram did not study global eutrophication; references 13, 14 and 16 are indirect and provide a large amount of unnecessary information (especially 13 and 16).
Line 62-71. The paragraph discusses well-known facts and is not interesting. Picoplankton does not relate to the topic of the article, especially in such unmodern and superficial aspect.
Line 72-81. The same is for cyanobacterial pigments: statement of well-known facts, not interesting and hard to read.
Line 100-119. This very important paragraph should be improved. Firstly, it concerns the image. To find this lake in the figure, I had to use the Internet and then again search for it on the map in the article. The image is poorly visualized, and the map of Estonia is emphasised for an unknown reason or perhaps for politicizing the situation. Such unclear images always reduce the significance of the publication.
Secondly, I would recommend placing this text at the beginning of the article, explaining that this is the fifth largest lake in Europe. Instead of an excessive review of the literature on cyanobacteria, you should add the historical aspect of ecological research in this particular hydrobiological object: anthropogenic pollution and history of cyanobacterial blooms.
To get this information, in addition to the article, I had to read the monograph “Lake Peipsi” published by Estonian Agricultural University in 2001…
Results
Fig 3. This is a very important image that needs the improvement: add a description of the cryptograms to the legend or a caption.
I did not found data of TP, TN, pH, water temperature, which were discussed in Results…..But however, this does not stop the authors from discussing them. Please provide this data to the text.
Fig 6. Mistake in abbreviation of microcystin E gene – mycE instead mcyE
Discussion
Line 228-230. The aim, stated by the authors, is fulfilled by other methods, which did not in this work, I do not recommend writing such way.
Line 235-240. Authors speculate authors the terms” toxic cyanobacteria”…very bored discussion...
Line 249. It started text about ecology of producing of toxic metabolites by cyanobacteria. But explanations are very weak…. The main idea of the Zapomelova's article is that Dolichospermum spp strains have a wide temperature range of growth. Also, water mixing is very important. Triggers of Dolichospermum spp blooming are low nitrates and absence of water mixing.
Line 272. The authors discuss the role of bioavailable phosphorus very hard, but do not provide references that cyanobacteria can independently release it (Guedes et al., 2019, DOI: 10.1016/j.hal.2019.03.006). especially Microcystis spp.
Line 319, discussion about variants. Often more variants and higher concentration of microcystins are found in more eutrophic waters, despite one producer ...for example, Lake Michigan waters (Bartlett et al., 2018). It would be more interesting than speculating about early detection and correlation from line 340 to line 346.
Material and methods
Study site - It would be better to provide coordinates of sampling sites instead description of Lake Peipsi.
Please, extract cyanobacterial cultures from HAMBI/UHCC culture collection to separate paragraph of Material and methods
Author Response
Dear Reviewer,
Thank you for giving us the opportunity to respond to your questions and improve our manuscript. We found the comments and suggestions very constructive and helpful.
Dear Reviewer,
Thank you for giving us the opportunity to respond to your questions and improve our manuscript. We found the comments and suggestions very constructive and helpful. Please find below our detailed responses to the review.
Introduction
First, the title and the structure of the Introduction are not good. The authors give a strange literature review, discussing well-known facts, and often give not very correct and indirect references. It would be better if this lake, the fifth largest in Europe, were more clearly emphasized:
the name should have no parentheses, and the ecosystem of this important transboundary object should be described not in the penultimate paragraph but at the beginning of the article. Highlighting the significance of the study of cyanobacteria in freshwater ecosystems does not require a detailed and partly unnecessary literature review on toxic blooms. Many references are incorrect. For example, Chorus and Bartram did not study global eutrophication; references 13, 14 and 16 are indirect and provide a large amount of unnecessary information (especially 13 and 16).
Authors: Thank you for this observation as it allows us to clarify our work. We shortened and restructured the introduction. We added a paragraph about the biggest transboundary lake in Europe in the introduction as suggested also by another reviewer. References in the MS have been controlled and parentheses have been removed as suggested.
Line 62-71. The paragraph discusses well-known facts and is not interesting. Picoplankton does not relate to the topic of the article, especially in such unmodern and superficial aspect.
Authors: Thank you for the comment. The introduction was shortened and this paragraph was omitted from the manuscript.
Line 72-81. The same is for cyanobacterial pigments: statement of well-known facts, not interesting and hard to read.
Authors: Thank you for the comment. The introduction was shortened and this paragraph was omitted from the manuscript.
Line 100-119. This very important paragraph should be improved. Firstly, it concerns the image. To find this lake in the figure, I had to use the Internet and then again search for it on the map in the article. The image is poorly visualized, and the map of Estonia is emphasised for an unknown reason or perhaps for politicizing the situation. Such unclear images always reduce the significance of the publication.
Authors: Thank you for the feedback, we do agree that the figure made for the first version of the manuscript was not the best. To achieve better clarity, Figure 1 has been replaced as recommended.
Secondly, I would recommend placing this text at the beginning of the article, explaining that this is the fifth largest lake in Europe. Instead of an excessive review of the literature on cyanobacteria, you should add the historical aspect of ecological research in this particular hydrobiological object: anthropogenic pollution and history of cyanobacterial blooms.
Authors: Thank you for your valuable comment. We revised the introduction accordingly, we added a paragraph about the biggest transboundary lake in Europe in the introduction as suggested also by another reviewer. Lines 87 – 100.
To get this information, in addition to the article, I had to read the monograph “Lake Peipsi” published by Estonian Agricultural University in 2001…
Authors: We added more comprehensive paragraph about the lake in the introduction. Lines 87 – 100.
Results
Fig 3. This is a very important image that needs the improvement: add a description of the cryptograms to the legend or a caption.
Authors: Thank you for the comment. Legend was modified accordingly. Basins and sampling points are now connected in the legend. Lines 157-159: Cyanobacterial community composition in different basins (sampling stations) of Lake Peipsi. Sampling points 2, 4, 5, 7, 10, 11, 38, 43, 56 and 91 are located in Lake Peipsi s.s.; sampling points 16 and 17 in Lake Lämmijärv and 22, 27, 51, 52 in Lake Pihkva.
I did not find data of TP, TN, pH, water temperature, which were discussed in Results…..But however, this does not stop the authors from discussing them. Please provide this data to the text.
Authors: Short paragraph and Table 1 containing the mentioned parameters has been added to the manuscript. (Lines 124-130)
Fig 6. A mistake in the abbreviation of microcystin E gene – mycE instead mcyE
Authors: Thank you for pointing this out, the misspelling is corrected on the figure.
Discussion
Line 228-230. The aim, stated by the authors, is fulfilled by other methods, which did not in this work, I do not recommend writing such way.
Authors: We modified the manuscript to make it more concise – Introduction was made more compact and the purpose of the study more clearly presented. We critically re-evaluated our aims and hypotheses (lines 114-121) and tried to keep the focus on them in all sections of the manuscript. Concluding remark were added to the end of Discussion. Lines 413 – 421: “To conclude, we demonstrated that even though generally the number of mcyE gene copies increased together with toxin concentration, mcyE copy number could not be used as a direct predictor of MC concentration in Lake Peipsi. Additionally, we showed that specific toxin variants were directly related to certain cyanobacterial genera. P. agardhii was significantly related with only [D-Asp3]MC-RR and other MC variants formed a close group with M. wesenbergii, M. aeruginosa in D. flos-aquae. Further, nitrate was the only nutrient related variable connected to MC quota per mcyE gene. Strong positive correlation between water temperature and MC quota per mcyE gene suggests that the warming trends might lead to more harmful cyanobacterial blooms in temperate shallow lakes.”
However, we are not truly sure what reviewer means “by other methods”.
Line 235-240. Authors speculate authors the terms” toxic cyanobacteria”…very bored discussion...
Authors: These sentences were deleted from the manuscript.
Line 249. It started text about the ecology of producing toxic metabolites by cyanobacteria. But explanations are very weak…. The main idea of Zapomelova's article is that Dolichospermum spp strains have a wide temperature range of growth. Also, water mixing is very important. Triggers of Dolichospermum spp blooming are low nitrates and absence of water mixing.
Authors: We agree, the water mixing is also very important and July is the most probable periods of low wind and warm weather leading to even stratification. These conditions can occur during periods of several weeks. Concentration of nitrates is low all-over summer period. Lines 274 – 277 were rephrased.
Line 272. The authors discuss the role of bioavailable phosphorus very hard but do not provide references that cyanobacteria can independently release it (Guedes et al., 2019, DOI: 10.1016/j.hal.2019.03.006). especially Microcystis spp.
Authors: Thank you for suggesting the interesting and valuable paper. Suggested reference added. Lines 293 - 295
Line 319, discussion about variants. Often more variants and a higher concentration of microcystins are found in more eutrophic waters, despite one producer ...for example, Lake Michigan waters (Bartlett et al., 2018). It would be more interesting than speculating about early detection and correlation from line 340 to line 346.
Authors: Thank you for the comment. The lead sentence to the paragraph was added on lines 332 – 335 and lines about the early correlation were deleted from Discussion.
Material and methods
Study site - It would be better to provide coordinates of sampling sites instead of a description of Lake Peipsi.
Authors: We agree, Table S1 with coordinates has been added as supplemental material
Please, extract cyanobacterial cultures from HAMBI/UHCC culture collection to separate paragraph of Material and methods
Authors: Thank you for your comment. Separate paragraph 5.6 about cultures has been added to Material and methods lines 373 – 375.
Reviewer 4 Report
Summary
The study sought to quantify the potential microcystin producers in a eutrophic lake in eastern Estonia using a combination of microscopy, qPCR, and LC/MS. Relationships were drawn between all potential microcytin producers and the mcyE copy number of three genera based upon Mantel test. Positive relationships were drawn between MC quota and temperature, but nitrate and phosphate driving biomass at different times of the year. The study represents a first published record (as far as this reviewer can tell) using qPCR to evaluate toxin potential in this lake.
Broad comments
Overall, the manuscript was well written and used some intriguing methods to evaluate the potential producers of MC in a large lake in eastern Estonia. I saw nothing fundamentally wrong with the study but had some suggestions on different ways to present the data and some minor comments. The authors should make efforts to add tables representing the underlying data to supplemental material as many of these are only presented as figures (e.g. counts for all taxa quantified, all LC/MS results, all environmental data mentioned (for instance SRP is not in Table A1), etc..).
Specific comments
For the correlation results, consider adding a figure using R’s corrplot (https://cran.r-project.org/web/packages/corrplot/vignettes/corrplot-intro.html) so that these results could be visualized? If not, could a table be produced to show the results of the analysis? Could be supplemental. Was there any multiple testing done to look for false discovery with the correlation calculations?
It would be nice to have all the environmental parameters mentioned in the text represented in Table A1. Also, LC/MS results should be represented in a table as well.
Line 121 – consider making a table of the microscopy results, could be supplemental. They are mentioned in the manuscript, but only three genera are represented in Fig 2, and other readers would be interested in these data.
Line 123 – add “were” between “cryptomonads” and “in”
Figure 3 – consider labeling which sampling points belong to which basin to assist readers in visualizing the main points of the figure which are highlighted in the text.
Figure 4 – Font is really small on the axis.
Line 162 – remove “concentration” after “microcystin”
Line 178 – I’m hesitant about this analysis, using mcyE abundance of three genera to describe a relationship with other potentially toxic species. The underlying data is abundance based upon microscopy, correct? It would be helpful to see the rest of the data (beyond the three genera that were tested for mcyE) to better evaluate it.
Figure 5 – Interesting that hypereutrophic Pihkva only is toxic once in Aug., also I’ve never seen a y-axis transformed like this, if it was transformed, should be mentioned in the legend.
Figure 6 – very hard to read the light green text
Figure 7 – consider using filled shapes to make it easier for the reader to see
Line 232 – CCC is used some places, CY community in others, please check and use one or the other throughout
Line 251 – Not measuring “toxicity” here, just toxin concentrations
Lines 297 to 302 – I haven’t read Tanner et al., but since the current study integrated across the water column, how well does it really compare?
Line 317 – how would this make it somewhat biased when we can look for so many variants in this day and age?
Line 379 – Horst et al calculated toxin quota as a function of biomass, so not directly comparable to your method, same with Wood et al. Part of the potential problem with the method employed by the authors is it assumes mcyE abundance is relatable to toxin concentration. There can be a host of environmental parameters that influence microcystin production and so it is not surprising that these two parameters might be anticorrelated.
Line 388 – maybe describe a bit more what makes these basins so hydrologically and morphometrically different, beyond trophic status
Line 404 – what did sampling point and time have to do with the volume that was collected?
Line 408 – how were samples stored on the boat? At -80C? If not, please state.
Line 421 – why was only a subset analyzed for MC?
Line 440 – add “(see below)” after “cultures” to guide the reader that you have more info on these procedures
Line 722 – hopefully this manuscript gets published before this one, if not it should be removed.
Author Response
Dear reviewer,
Thank you for pointing on the issues which should be addressed in order to improve our MS. We found the comments and suggestions very constructive and helpful.
Summary
The study sought to quantify the potential microcystin producers in a eutrophic lake in eastern Estonia using a combination of microscopy, qPCR, and LC/MS. Relationships were drawn between all potential microcystin producers and the mcyE copy number of three genera based upon Mantel test. Positive relationships were drawn between MC quota and temperature, but nitrate and phosphate driving biomass at different times of the year. The study represents a first published record (as far as this reviewer can tell) using qPCR to evaluate toxin potential in this lake.
Broad comments
Overall, the manuscript was well written and used some intriguing methods to evaluate the potential producers of MC in a large lake in eastern Estonia. I saw nothing fundamentally wrong with the study but had some suggestions on different ways to present the data and some minor comments. The authors should make efforts to add tables representing the underlying data to supplemental material as many of these are only presented as figures (e.g. counts for all taxa quantified, all LC/MS results, all environmental data mentioned (for instance SRP is not in Table A1), etc..).
Dear reviewer,
Thank you for pointing on issues which should be addressed in order to improve our MS. Please refer to the detailed responses below.
Specific comments
For the correlation results, consider adding a figure using R’s corrplot (https://cran.r-project.org/web/packages/corrplot/vignettes/corrplot-intro.html) so that these results could be visualized? If not, could a table be produced to show the results of the analysis? Could be supplemental. Was there any multiple testing done to look for false discovery with the correlation calculations?
Authors: Thank you for your suggestions, we were considering to add the results of the correlation analysis in the supplements. But we realised that additional plots did not add much for the manuscript. In these cases we did not use any multiple testing.
It would be nice to have all the environmental parameters mentioned in the text represented in Table A1. Also, LC/MS results should be represented in a table as well.
Authors: Thank you! Data mentioned in the text have been added to Table 1 and Table A1.
Line 121 – consider making a table of the microscopy results, could be supplemental. They are mentioned in the manuscript, but only three genera are represented in Fig 2, and other readers would be interested in these data.
Authors: Thank you for the good suggestion, the table with microscopy results was added as a supplemental file.
Line 123 – add “were” between “cryptomonads” and “in”
Authors: „were“ has been added.
Figure 3 – consider labelling which sampling points belong to which basin to assist readers in visualizing the main points of the figure which are highlighted in the text.
Authors: Thank you for the comment. Legend was modified accordingly. Basins and sampling points are now connected in the legend. Lines 157-159: Cyanobacterial community composition in different basins (sampling stations) of Lake Peipsi. Sampling points 2, 4, 5, 7, 10, 11, 38, 43, 56 and 91 are located in Lake Peipsi s.s.; sampling points 16 and 17 in Lake Lämmijärv and 22, 27, 51, 52 in Lake Pihkva.
Figure 4 – Font is really small on the axis.
Authors: Font size was increased as suggested.
Line 162 – remove “concentration” after “microcystin”
Authors: „concentration“ has been removed.
Line 178 – I’m hesitant about this analysis, using mcyE abundance of three genera to describe a relationship with other potentially toxic species. The underlying data is abundance based upon microscopy, correct? It would be helpful to see the rest of the data (beyond the three genera that were tested for mcyE) to better evaluate it.
Authors: The reason why we analyse the data in such a way is that we use the primers which are specific to these three genera, no other genera are covered with selected primers. Clarifying comment in M&M (lines 490-492).
Figure 5 – Interesting that hypereutrophic Pihkva only is toxic once in Aug., also I’ve never seen a y-axis transformed like this, if it was transformed, should be mentioned in the legend.
Authors: On y-axis scale square root transformation was applied. Clarifying remark was added to the legend of Figure 5. Lake Pihkva (Russia) was only sampled in August during the Estonian–Russian joint expeditions, other times the access to Russian territory was not possible.
Figure 6 – very hard to read the light green text
Authors: Thank you! Figure 6 has been corrected as suggested, colours on the figure have been changed.
Figure 7 – consider using filled shapes to make it easier for the reader to see
Authors: Thank you! Figure 7 has been corrected as suggested. Filled shapes are now used on Figure 7.
Line 232 – CCC has used some places, CY community in others, please check and use one or the other throughout
Authors: Thank you, now CCC is being used throughout the manuscript
Line 251 – Not measuring “toxicity” here, just toxin concentrations
Authors: The sentence was changed as follows „Another aim of this study was to use molecular markers to identify and quantify the potential microcystin producers in the basins of Lake Peipsi. For that purpose, genus-specific qPCR was used. Lines 262 - 263
Lines 297 to 302 – I haven’t read Tanner et al., but since the current study integrated across the water column, how well does it really compare?
Authors: Lake Peipsi is a large and shallow wind-exposed lake, which water column is continuously mixed due to wind activity. In the study of Tanner et al., the open water area, samples were collected from the upper layer (until 1 m from the surface) but there was no surface scum recorded. In our study, also no surface scum was recorded in the open water area and temperature and oxygen profiles indicated that the water column was well mixed. Based on the fact that it is a shallow well mixed layer, and both studies had no scum during sampling we believe that they had quite similar conditions during both studies and thus we can make comparisons between these two studies.
Line 317 – how would this make it somewhat biased when we can look for so many variants in this day and age?
Authors: Thank you very much for pointing this out. Yes, we do agree, that at the present age, there is a possibility to analyse a high number of variants. We removed this outdated explanation.
Line 379 – Horst et al calculated toxin quota as a function of biomass, so not directly comparable to your method, same with Wood et al. Part of the potential problem with the method employed by the authors is it assumes mcyE abundance is relatable to toxin concentration. There can be a host of environmental parameters that influence microcystin production and so it is not surprising that these two parameters might be anticorrelated.
Authors: Thank you for pointing this out. We made clear when quota per biomass and per gene abundance is used in the revised version of the manuscript. Gene abundance is often used as a proxy for toxin production even in recent literature. We assume that biomass of cyanobacteria in general is even less reliable predictor for toxin concentration and our verdict is that the predictive power is increasing from measuring the absolute abundance of toxin production genes towards to quota per genes and perhaps is even increased by measuring gene expression level. However, even specific mRNA levels are not 100% predictive to protein activity and therefore for enzymatic reactions behind the toxin production rates of cyanobacteria.
Line 388 – maybe describe a bit more what makes these basins so hydrologically and morphometrically different, beyond the trophic status
Authors: This section has been deleted from Material and methods section and has been added to the Introduction according to suggestions by other reviewers.
Line 404 – what did sampling point and time have to do with the volume that was collected?
Authors: At the beginning of the vegetation period, water is more clear and the filters do not clog up so easily, also the same thing with different sampling points, in more eutrophic parts of the lake, filters are clogging with the particles in the water more easily and thus smaller volume of water can be processed.
Line 408 – how were samples stored on the boat? At -80C? If not, please state.
Authors: Thank you for pointing this out. The explanation was added in lines 434 – 435.
Line 421 – why was only a subset analyzed for MC?
Authors: For MC analyses, unfortunately, we were depending on the collaboration with the foreign laboratory and due to technical reasons we were able to collect samples for MC analyses after the study had already started. Therefore, we have fewer samples for MC analyses.
Line 440 – add “(see below)” after “cultures” to guide the reader that you have more info on these procedures
Authors: Short separate paragraph 5.6 (as suggested also by another reviewer) was added before DNA extraction paragraph.
Line 722 – hopefully this manuscript gets published before this one, if not it should be removed.
Authors: Thank you for the comment, unfortunately, in the guidelines for the authors, there is a note that "Data not shown" should be avoided and unpublished data should be cited in the text and a reference should be added in the References section. Unpublished work should be cited as follows. Author 1, A.B.; Author 2, C. Title of Unpublished Work. status (unpublished; manuscript in preparation).